

# Mitigation of bias sources for atmospheric temperature and humidity in the mobile Weather & Aerosol Raman Lidar (WALI)

**Julien Totems, Patrick Chazette and Alexandre Baron**

[1]{Laboratoire des Sciences du Climat et de l'Environnement, CEA, Gif-sur-Yvette, France}

Correspondence to: J. Totems (julien.totems@cea.fr)

**Abstract**

Lidars using vibrational and rotational Raman scattering to continuously monitor both the water vapor and temperature profiles in the low and middle troposphere offer enticing perspectives for applications in weather prediction and studies of aerosol/cloud/water vapor interactions by deriving simultaneously relative humidity and atmospheric optical properties. Several heavy systems exist in European laboratories but only recently have they been downsized and ruggedized for deployment in the field. In this paper, we describe in detail the technical choices made during the design and calibration of the new Raman channels for the mobile Weather and Aerosol Lidar (WALI), going over the important sources of bias and uncertainty on the water vapor & temperature profiles stemming from the different optical elements of the instrument. For the first time, the impacts of interference filters and non-common-path differences between Raman channels, and their mitigation, are particularly investigated, using horizontal shots in a homogenous atmosphere. For temperature, the magnitude of the highlighted biases can be much larger than the targeted absolute accuracy of 1°C defined by the WMO. Measurement errors are quantified using simulations and a number of radiosoundings launched close to the laboratory.

## 1   Introduction

Atmospheric temperature and humidity in the low atmosphere are together essential to comprehend weather phenomena and their evolution in a changing climate. Through the effect of relative humidity on aerosol hygroscopicity and cloud formation, they also influence the radiative balance of the Earth, generating the largest uncertainties in climate projections (IPCC, 2013). For both weather and climate prediction, observation means have evolved tremendously, notably with satellite retrievals of moisture and temperature routinely assimilated in numerical models. Yet remote-sensing techniques from spaceborne missions have difficulties probing the



lower troposphere below 2-3 km in altitude, and have vertical resolutions that are too low,
greater than 1 km in the lower troposphere (e.g. Prunet et al., 1998; Crevoisier et al., 2014).
They are thus unable to resolve temperature inversions and thin dry/humid air masses (e.g.
Chazette et al., 2014; Hammann et al., 2015; Totems et al., 2019). Providing complementary
profiles of the important thermodynamic variables in the first kilometres of the atmosphere,
where most of the water vapour and temperature vertical variability is confined, is of paramount
importance for both weather forecast and reducing aerosol-induced uncertainty on climate
models (Wulfmeyer et al., 2015).
Given their capacity for continuous, well-resolved and precise temperature measurements in
the lower troposphere, Vibrational Raman (VR) and Rotational Raman (RR) lidars have
emerged as adequate tools in this endeavour. Water vapor profilers are now well-established
(from Whiteman et al., 1992 to e.g. Dinoev et al., 2013), whereas temperature profilers have
recently become more widespread and powerful (from Cooney, 1972 and Vaughan et al., 1993
to e.g. Weng et al., 2018 or Martucci et al., 2021). Without tackling turbulence-scale resolution
which is the prerogative of heavier systems like the Raman lidars of the University of
Hohenheim (Behrendt et al., 2015), the University of Basilicata (Di Girolamo et al., 2017) or
ARTHUS (Atmospheric Raman Temperature and Humidity Sounder, Lange et al., 2019), there
is a need for field-deployable instruments capable of fulfilling the breakthrough requirements
set by the World Meteorological Organization in terms of accuracy on atmospheric temperature
and humidity in the low troposphere (WMO, 2017). Lidar profiles have proven beneficial for
both numerical weather prediction (NWP) models (e.g. Adam et al., 2016; Fourrié et al., 2019),
the study of dynamic processes in the planetary boundary layer (PBL) (e.g. Behrendt et al.,
2015) or interactions between water vapor and aerosols (e.g. Navas-Guzmán et al., 2019). But
to obtain the absolute accuracies demanded here, especially that of 1°C or less on temperature,
the required accuracy on the lidar channel ratios and their calibration is extremely stringent,
and the sources of bias seldom discussed in the literature (Behrendt and Reichardt, 2000;
Simeonov et al., 1999; Whiteman et al., 2012).
Within the European lidar landscape, WALI (Weather and Aeorosol Lidar) is a seasoned mobile
Rayleigh-Mie-Raman system, eyesafe at 355 nm, first deployed during the HyMeX
international field campaign and subsequently ChArMEx and PARCS, for aerosol and water
vapor profiling (resp. Hydrological cycle in the Mediterranean eXperiment, Chemistry and
Aerosol in the Mediterranean Experiment, Pollution in the Arctic System; Chazette et al.,
2014b, 2018; Totems et al., 2019; Totems and Chazette, 2016). In its latest evolution, the VR
channels have been replaced by a Newton reflector and a polychromator also including RR





channels for temperature profiling. On this occasion, we have established that biases due to
various sources, in particular from the dependency of spectral filtering on the angle of
incidence, detector non-uniformities and other non-common-path differences between Raman
channels, may be several times greater than the requirements if left unchecked. Correctible as
they are by measuring the ratios of overlap factors on the individual channels, these effects are
not reported in the literature of lidar temperature measurements. However, they were bound to
appear given the physical characteristics of the systems mentioned hereabove.
The aims of this paper are: i) to compile for the first time the sources of bias that must be
considered and mitigated when using a Raman lidar to profile atmospheric temperature and
humidity, ii) to validate WALI as a dependable profiler deployable for field campaigns,
satisfying the requirements set by the WMO.
The theory of the Raman lidar retrieval of the atmospheric temperature and WVMR, the error
budget on these parameters, and the known sources of bias are recalled in section 2, as well as
the principle and limitations of the overlap measurement method. In section 3, after
summarizing the characteristics of WALI, we propose a sequential review of the components
of the lidar chain, characterizing and mitigating the error sources. The results of a calibration
and qualification experiment using radiosondes follow in section 4. A conclusion and outlooks
are presented in section 5.

## 2    Theoretical considerations

### 2.1    Raman lidar retrieval of humidity and temperature

We will introduce notations by briefly recalling the theory of the retrieval of water vapor content
and temperature by the Raman lidar technique; the complete theory has been extensively
derived before, by Whiteman et al. (1992) and Behrendt (2005) respectively, among others.
The vertical profiles of water vapor mixing ratio (WVMR) $r_{H2O}$ and temperature $T$ are
calculated from the ratios of the $H_2O$ / $N_2$-vibrational Raman (VR) channels and the RR2 (high-
J number) / RR1 (low-J number) rotational Raman (RR) channels, respectively:

$$R(z) = \frac{S_{H_2O}(z)}{S_{N_2}(z)} \qquad (1)$$

$$Q(z) = \frac{S_{RR2}(z)}{S_{RR1}(z)} \qquad (2)$$





Signals $S_j(z)$ of Raman channels $j$ have all been previously averaged over the required altitude
and time to improve the signal to noise ratio (SNR), and corrected for i) electronic baseline
variations by subtracting a baseline recorded every few profiles with detector (photomultiplier
tube, PMT) gain set to zero, ii) the sky background mean value assessed on pre-trigger or post-
signal samples, iii) PMT gain variations (allowed on the VR channels to optimize daytime
dynamic range, eg. Chazette et al. (2014b)), iv) known leakage of the elastic return in the RR
filters (Behrendt and Reichardt, 2000). $S_j(z)$ are thus expressed as:

$$S_j(z) = \frac{1}{G_j(U_j)}\left(S_{j,raw}(z) - \hat{L}_j(z) - \hat{B}_j\right) - \hat{\varepsilon}_j S_{elas}(z) \tag{3}$$

where $G_j$ is the channel gain controlled by PMT voltage $U_j$, $S_{j,raw}$ is the raw lidar signal, $\hat{L}_j$ is
the estimated baseline, $\hat{B}_j$ is the estimated sky background parasitic signal, $\hat{\varepsilon}_j$ is the estimated
residual transmittance of the emitted laser wavelength through the interference filter (IF) of
Raman channel $j$ compared to the elastic channel, and $S_{elas}$ is the elastic signal.
Both $R$ and $Q$ must then also be corrected from the difference of atmospheric transmission
between the two Raman channels and the ratio of overlap factors:

$$R'(z) = \frac{\exp(\Delta\tau(z))}{\widehat{OR_R}(z)} R(z) \tag{4}$$

$$Q'(z) = \frac{1}{\widehat{OR_Q}(z)} Q(z) \tag{5}$$

where $\Delta\tau(z)$ is the difference of optical thickness from the lidar until range z observed between
the wavelengths of the two VR channels, and where $\widehat{OR_R}(z)$ and $\widehat{OR_Q}(z)$ are the estimated
ratios of the overlap factors of the two VR / RR channels respectively (expressed in section
2.4). With an emitted wavelength at 355 nm, $\Delta\tau(z)$ between 387 and 407 nm seldom produces
deviations above 5%, and can be efficiently estimated using an average atmospheric density
profile for molecular optical thickness and the $N_2$-Raman channel itself for aerosol optical
thickness (e.g. Whiteman, 2003).
The WVMR is simply proportional to the VR scattering ratio between $H_2O$ and $N_2$, since the
latter can be considered with a constant mixing ratio in the troposphere and stratosphere. The
temperature is retrieved from the more complex dependency of the RR scattering cross sections
between the two channels RR1 and RR2. The respective estimates $\hat{r}_{H_2O}$ and $\hat{T}$ are obtained,
after calibration, by:





$$\hat{r}_{H_2O}(z) = \hat{K}R'(z) \tag{6}$$

$$\hat{T}(z) = \hat{f}^{-1}\big(Q'(z)\big) \tag{7}$$

where $\hat{K}$ is the estimate of the calibration coefficient for WVMR combining all instrumental
constants. Calibration function $\hat{f}$ is the estimate of the temperature dependency of the ratio of
RR cross-sections. It takes into account the instrumental constants of the two RR channels. We
take the model previously selected for operational purposes by Behrendt (2005):

$$Q' = f(T) = \exp\left(a + \frac{b}{T} + \frac{c}{T^2}\right) \tag{8}$$

with $a$, $b$, $c$ the coefficients of a polynomial regression of $\ln(Q')$ as a function of $1/T$. $\hat{K}$ and $\hat{f}$
are obtained by confronting lidar profiles of $R'$ and $Q'$ with collocated in-situ measurements of
$r_{H2O}$ and $T$ (e.g. from a radiosounding), aiming for a wide range of values for a better constraint
on the calibration.

## 2.2  Simple error budget

In this section, we will make a first assessment of the acceptable error on $R$ and $Q$ starting from
the accuracy requirements for WVMR and temperature profiles, which ensue from each
scientific need, as compiled by Wulfmeyer et al. (2015) for key applications. Monitoring,
verification (e.g. model qualification or calibration/validation of satellites) and data
assimilation purposes can be adequately addressed by a profiler capable of i) <5% noise error
and <2-5% bias for water vapor, ii) <1°C noise error and <0.2-0.5°C bias for temperature. In a
simple error budget, we can use requirements of $\left(\frac{\Delta r_{H_2O}}{r_{H_2O}}\right)_{max} = 5\%$ for WVMR, and
$\Delta T_{max} = 1°C$ for temperature, to give a first idea of the different expectations for the
performance of a VR/RR lidar.
Eqs. (4-8) allow to derive constraints on the acceptable relative error on the corrected lidar
observables $R'$ and $Q'$, for either random noise or bias, as:

$$\left(\frac{\Delta R'}{R'}\right)_{max} = \left(\frac{\Delta r_{H_2O}}{r_{H_2O}}\right)_{max} \tag{9}$$

$$\left(\frac{\Delta Q'}{Q'}\right)_{max} = \frac{dQ'/dT}{Q'}\Delta T_{max} \tag{10}$$





The relative error on $R$ is equal to the constraint on WVMR, i.e. 5%. An assessment of the
relative error on $Q$ is performed considering the RR filter parameters given in Table 2 (section
3) to yield the following numerical application: around $T_0 = 0°C$, $Q'(T_0) = 0.44$ and $dQ'/dT(T_0)$
$= +0.35/100°C$, so that: $\left(\frac{\Delta Q'}{Q'}\right)_{max} = 0.79\% \, \Delta T_{max}(°C)$.
Table 1. Summary of accuracy requirements from Wulfmeyer et al. (2015) and corresponding
constraints on ratios $R'$ and $Q'$. Resulting errors on relative humidity $RH$ at 0°C and 50%RH.

| Parameter | Random error | Systematic error (bias) |
|---|---|---|
| $r_{H2O}$ | <5% relative | <2-5% relative |
| $T$ | <1°C | <0.2-0.5°C |
| $R'$ | <5% i.e. $SNR > 20$ | <2-5% |
| $Q'$ | <0.8% at 0°C i.e. $SNR > 125$ | <0.12-0.4% at 0°C |
| $RH$ | 4.3%RH | 1.2-2.9%RH |
| | at $T = 0°C$, $RH = 50\%$ | at $T = 0°C$, $RH = 50\%$ |


The results, summarized in Table 1, have very important implications. Typically, $Q'$ must be 6
to 10 times more accurate than $R'$ to deliver meaningful results in terms of temperature. Raman
cross-sections being larger for the RR channels than for the H$_2$O VR channel, the main
difficulties shift from constraints linked to signal-to-noise ratio (SNR) to also encompass strong
constraints linked to instrumental biases. SNR as used in Table 1 is defined on $R$ and $Q$ at the
final resolution, and is calculated from the individual signal variances (including laser & sky-
background photon noise, detection noise), as:

$$SNR_R = \left(\frac{\text{var}(S_{N_2})}{\langle S_{N_2}\rangle^2} + \frac{\text{var}(S_{H_2O})}{\langle S_{H_2O}\rangle^2}\right)^{-\frac{1}{2}} \qquad (11)$$

$$SNR_Q = \left(\frac{\text{var}(S_{RR1})}{\langle S_{RR1}\rangle^2} + \frac{\text{var}(S_{RR2})}{\langle S_{RR2}\rangle^2}\right)^{-\frac{1}{2}} \qquad (12)$$

$SNR_R$, typically limited by the H$_2$O channel, must be above ~20 and $SNR_Q$ must be above ~125
to satisfy the requirements given above. Such high values can be reached by increasing the laser
power and pulse repetition frequency (PRF), or enlarging the integration over altitude and time,
as SNR is usually magnified by the square roots of the energy and number of averaged samples.
However, limits on the latter are also set by Wulfmeyer et al. (2015) for the same applications;





integration range $\Delta z$ should be below 100 m in the PBL and 300 m in the lower free troposphere,
whereas an integration time $\Delta t$ between 15 (assimilation and verification) and 60 min
(monitoring) is required.
We derive the errors expected on RH given those on temperature and WVMR at the bottom of
Table 1. Here and in the following, %RH denote absolute percentage units on RH, whereas %
denote relative errors. Relative humidity is derived as a function of atmospheric pressure,
temperature and WVMR, using standard empirical relationships for the water vapor saturation
pressure. Here, we use the Buck equation (Buck, 1981), which is accurate within 0.2% between
−40°C and +100°C:

$$P_{wv,sat} = 6.1121 \frac{T}{T + 257.14°C} \exp\left(18.678 - \frac{T}{T + 234.5°C}\right) \tag{13}$$

$$RH = \frac{P}{P_{wv,sat}} \frac{r_{H_2O}}{r_{H_2O} + 621.991 \text{ g kg}^{-1}} \tag{14}$$

with $P$ pressure and $P_{wv,sat}$ the water vapor saturation pressure in hPa, $T$ temperature in °C.

## 2.3   Sources of bias

Biases arising from inaccurate measurement of any of the estimated factors of Eqs. (3-7), or
from a variation after that measurement due to instabilities in the instrument, must also be
smaller than the aforementioned values of 2-5% for WVMR and 0.12-0.4%, an especially
difficult goal to reach for temperature. Their impact must be mitigated either by careful design
or by precise estimation.
The expected (i.e. noiseless) values of $R$ and $Q$ can be detailed as:

$$\overline{R(z)} = \frac{O_{H_2O}(z)}{O_{N_2}(z)} \frac{K_{H_2O}}{K_{N_2}} \frac{\sigma_{H_2O}}{\sigma_{N_2}} r_{H_2O}(z) \tag{15}$$

$$\overline{Q(z)} = \frac{O_{RR2}(z)}{O_{RR1}(z)} \frac{K_2 \sigma_{RR2}(T(z))}{K_1 \sigma_{RR1}(T(z))} \tag{16}$$

with $\bar{x}$ denoting the expected value of variable $x$, $K_j$ and $O_j(z)$ the instrumental constant and
overlap factor of channel $j$, respectively. To simplify our discussion, we choose to incorporate
any deviation that affects the ratios without a range-dependence into the instrumental constant
ratio, and any deviation with a range-dependence into the overlap ratio.
As previously explained, the impact of deviations on variables in Eq. (15) remains tolerable
below a few percent, but for the distinctly more constrained temperature retrieval, the variables
in Eq. (16) are affected by the following effects that directly induce significant bias:
• Laser wavelength drift or filter central wavelength (CWL) drift with temperature both
affect the ratios indiscriminately with range. By simulating the variation of Q with the
WALI filter parameters (section 3), we find a large impact of a wavelength drift $\Delta\lambda$
(measured between the laser on one side and both interference filters on the other side):
$dQ/Q\,/d\lambda \approx -0.26\;\mathrm{pm}^{-1}$ and $\Delta T \approx -0.34°\mathrm{C\;pm}^{-1}\,\Delta\lambda$, meaning just 3 pm drift in
either filter or laser wavelengths can lead to biases above 1°C. That is one of the reasons
why the laser must be frequency-stabilized. Also, IFs subjected to fluctuations of local
temperature are known to experience CWL drifts; for WALI's filters manufactured by
Materion, this amounts to 1.28 pm °C⁻¹ (value given by the manufacturer after their
material dilation simulation). The temperature of the polychromator must th us be kept
stable within 1°C for this bias to become negligible.
• Filter CWL variation with angle of incidence (AOI) on the IF generates a channel
transmittance variation which is range-dependent, and different for each filter. Indeed,
this variation $\Delta CWL$ is approached by (e.g. Hayden Smith and Smith, 1990):

$$\Delta CWL(\theta') \approx CWL\,\frac{\theta'^2}{2n_{eff}^2} \qquad (17)$$

where $CWL$ is the filter central wavelength, $\theta'$ is the angle of incidence on the filter
(assumed small), and $n_{eff}$ is the effective index of the filter. For the RR1 filter ($n_{eff} =$
1.62), we obtain as much as $\Delta CWL(\theta') \approx 43\;\mathrm{pm}\;\theta'(°)²$. The problem stems from the
fact that because the filter is in the pupil plane, after collimation of the received beam,
each angle of incidence corresponds to a different point in the focal plane of the receiver,
which in turns corresponds to a field angle $\theta$ of the lidar, as seen on Figure 1 a). Aperture
number conservation across the receiving optical system imposes

$$\theta' = \frac{f}{f'}\theta > \frac{D_{rec}}{D_{IF}}\theta \qquad (18)$$

where $f$ and $f'$ are the receiver/recollimation focal lengths, $D_{rec}$ and $D_{IF}$ are the receiver
and IF diameters. For a 150-mm diameter receiver using a 1-inch diameter (22 mm clear
aperture) IF, we obtain at least $\theta' = 0.39°$ for a $\theta = 1$ mrad field angle, producing
$\Delta CWL(\theta') \approx 6.6\;pm$ and already $\Delta T \approx 2,2°\mathrm{C}$. Note that the impact gets



proportionately larger with the diameter of the receiver. Because the optical path of each
channel is independently aligned, this always induces different overlap factors even
when sharing the same telescope. This large effect must be calibrated and corrected, yet
its impact was never discussed before in the RR lidar literature, despite being three times
as large in other systems with 450 mm receivers. This impact can be mitigated by
attacking the filters at normal incidence, where the derivative of CWL as a function of
AOI (see Eq. (17)) is minimal.

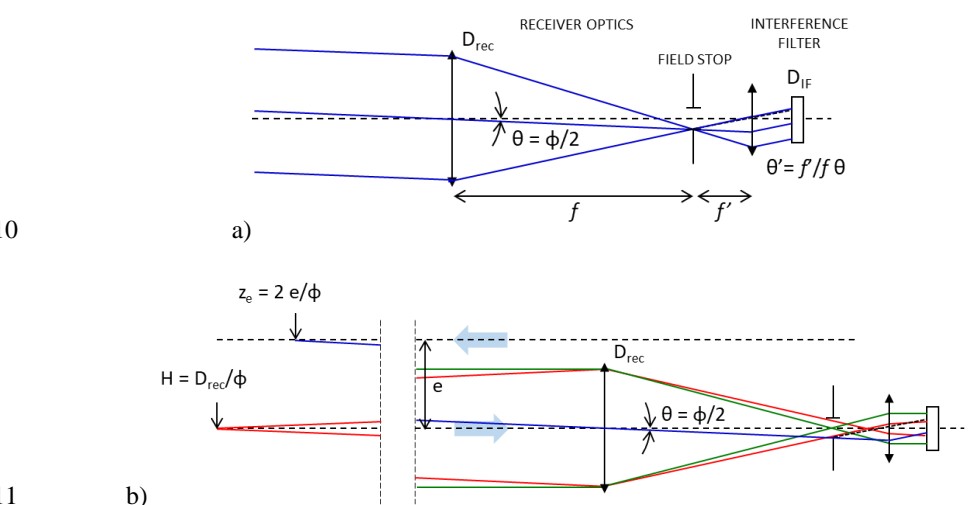

a)
b)
Figure 1. a) Definition of useful parameters for field angle $\theta$ and filter angle of incidence
$\theta'$ calculations. $f$: receiver focal length, $D_{rec}$: receiver diameter, $\phi$: full lidar field-of-
view, $f'$: collimation focal length, $D_{IF}$: IF diameter.  b) Definition of metrics for overlap
calculations. $e$: emitter-receiver separation, $H$: hyperfocal distance, $z_e$: entry distance of
laser into field-of-view.
•  Detector response non-uniformity up to $\pm 12\%$, both as a function of impact point on the
active surface and of angle of incidence, is now specified on the cathodes of PMTs used
at 400 nm wavelength (Hamamatsu (2007), Section 4.3.3). The amplitude was found to
be much larger by Simeonov et al. (1999), with significant impact. This effect has been
bluntly limited in all our lidars by putting the cathode plane as far as possible before the
focal plane, while still avoiding vignetting. It can still be responsible for differences of
overlap factors between channels.
•  Uncalibrated PMT gain or digitizer baseline variations will of course induce bias in the
channel system constants. We will see how to mitigate these effects.





• Slight variations of overlap or channel transmittance after calibration will be directly
responsible for bias. In the next sub-section, we discuss how they can appear.

## 2.4 Overlap measurement with horizontal shots and limitations

Range-dependent biases influence the lower part of the profiles just like overlap factors, at
varying distances from the emitter depending on both the quality of the alignments and
characteristics of the receiving optics. Two methods are used in the literature to estimate the
overlap factors of a Raman lidar: i) an iterative Klett inversion of elastic and Raman channels
sharing the same telescope is easy to achieve (Wandinger and Ansmann, 2002) but inefficient
when non-common path errors are involved, whereas ii) the method of aiming the lidar
horizontally (e.g. Sicard et al., 2002; Chazette and Totems, 2017) is sometimes impractical but
more direct and yields more accurate results in an horizontally homogeneous atmosphere over
a range of 1 to 2 km. In the context of RR measurements, it is necessary to implement the latter,
and also to measure the ratios of overlap factors, rather than the overlap factors themselves,
thus avoiding errors due to an imprecise estimation of atmospheric extinction.
Considering a horizontal line of sight in a supposedly homogeneous atmosphere, the expected
values of ratios $R$ and $Q$ can be expressed as:

$$\overline{R(z)} = R(z_\infty) \frac{O_{H_2O}(z)}{O_{N_2}(z)} \exp(-\Delta\alpha \cdot z) \tag{19}$$

$$\overline{Q(z)} = Q(z_\infty) \frac{O_{RR2}(z)}{O_{RR1}(z)} \tag{20}$$

where $R(z_\infty)$ and $Q(z_\infty)$ are the values observed when all overlap factors have become constant
at a sufficiently large range from the lidar, noted $z_\infty$, after which variations of the optical path
inside the reception channels become negligible. $\Delta\alpha = \alpha(407nm) - \alpha(387nm)$ is the difference
of atmospheric extinction between the two VR wavelengths.
To evaluate $z_\infty$, we introduce in Figure 1 b) parameters that characterize the overlap of a paraxial
or coaxial lidar (e.g. Kuze et al., 1998): i) $z_e = 2e/\phi$ at which the emitted laser beam located at
distance $e$ from the receiver axis enters the field of view, whose full size is $\phi$; $z_e$ is null for a
coaxial system ii) $H = D_{rec}/\phi$, the so-called hyperfocal distance, minimum range from which
the beam originating from a point still fully enters the field stop; iii) $H_{IF} = 2D_{rec}f/f'\theta'_{max}$, that
we might call the filter hyperfocal distance, similarly to the former, the minimum range from
which the image of a point does not exceed $\theta'_{max}$, the AOI on the IF that significantly changes
its transmittance. $z_\infty$ is above the maximum of those three, which is usually $H_{IF}$. If we use for





$\theta'_{max}$ the AOI value causing 1°C bias on temperature per Eq. (10) and Eq. (15), we find: $z_\infty > H_{IF}$
= 780 m. Note that $z_\infty$ can reach several km with misaligned filters.
If for instance the lidar can be mounted on a rotating platform capable of aiming horizontally,
the overlap ratios can be estimated with suitable precision by averaging over time and range
(and correcting for differential of extinction on the VR ratio):

$$\widehat{OR_R}(z) = \frac{R(z)}{R(z_\infty)} \exp(\Delta\alpha \cdot z) \tag{21}$$

$$\widehat{OR_Q}(z) = \frac{Q(z)}{Q(z_\infty)} \tag{22}$$

However, assumptions are made for this estimation, namely:
• As explained above, the atmosphere is assumed to be homogeneous in WVMR and
temperature (down to <0.5°C) up until $z_\infty$, whereas the overlap ratios must be constant
(down to <0.4%) after $z_\infty$. Also, the maximum range (with sufficient SNR) of the lidar
must exceed $z_\infty$, implying nighttime measurements for the Raman channels. Therefore,
the effects generating overlap variation after a few hundred meters must be prevented.
• The lidar is assumed to retain the exact same overlap functions when aiming
horizontally and vertically. Considering a field of view around 1 mrad, the stability of
the emission and reception optical paths must be better than ~10 µrad between these
two positions. This is feasible for a small refractor but difficult for a Raman system such
as WALI, with a heavy laser and large reflector.
These difficulties make it extremely challenging to estimate the overlap ratios with an accuracy
better than a few percent. This is enough for the WVMR, but we find that a correction must be
applied by comparing with in-situ sounding for temperature measurements by Raman lidar.
**3    Implementation and bias mitigation on the WALI system**
In this section, we describe the WALI instrument from the emitter to the reception channels,
characterizing the critical elements in the framework of WVMR and temperature
measurements. The system has evolved from its previous implementation described in Totems
et al. (2019), by adding RR channels and a fibered telescope receiver. A global diagram
presenting the main lidar sub-systems is shown in Figure 2, and a summary of its characteristics
is given in Table 2.

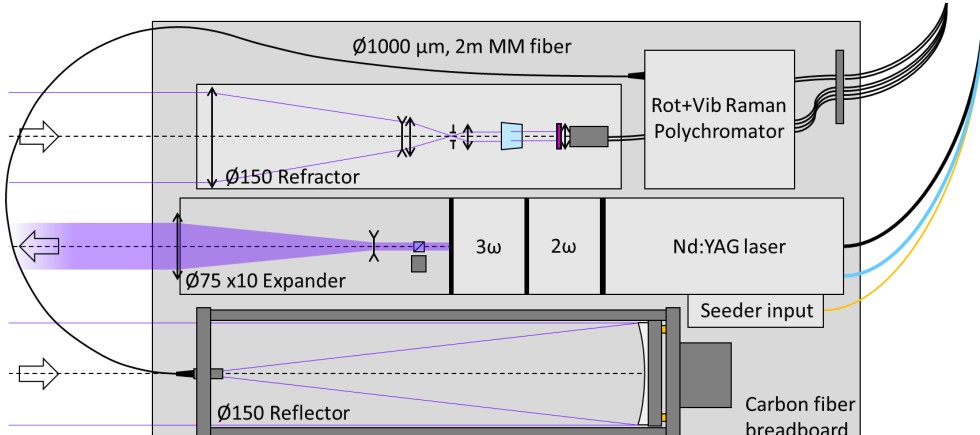


Figure 2. Global diagram of the lidar system. The main sub-systems are: the emitter (center),
the elastic receiver using a refractor (top), the Raman receiver using a fibered parabolic reflector
(bottom), and a separate, thermally stabilized polychromator (upper right). See Figure 5 for the
detail of the polychromator design.
Its main features are a single rotatable platform (lightweight carbon fiber breadboard by
CarbonVision GmbH) carrying both its emission and reception paths, a 150-mm refractor for
the elastic channels (for aerosol studies), and a 150-mm diameter parabolic fibered reflector for
all the Raman channels. The separation of the four Raman channels takes place in a deported
polychromator set in a thermally controlled enclosure, fed by the optical fiber. Fiber optics are
also known to partly scramble the input illumination, minimizing the range-dependance of filter
transmittance for the different Raman channels. The output signals from the photomultiplier
tubes (PMTs) in the polychromator are digitized by a NI$^{TM}$ PXI system (not shown).
Table 2. WALI instrument characteristics summary (PRF: pulse repetition frequency, FOV:
field of view, CWL: central wavelength in vacuum, FWHM: full width at half-maximum, OOB:
out-of-band blocking specification, OD: optical density)

| Emitter | Laser | Lumibird$^{TM}$ Q-Smart 450 SLM, tripled Nd:YAG, frequency stabilized $\lambda_{laser}$ = 354.725 nm in vacuum, $E_p$ = 100 mJ, PRF = 20 Hz. |
|---|---|---|
| | Optics | High-power polarizing beamsplitter and 10x beam expander Output beam diameter: 65 mm, Em/Rec separation: 200 mm |
| Elastic | Optics | Ø150 mm F/2 UV fused-silica refractor |
| receiver | Spatial filter | 0.67 x 2 mrad FOV |
| | Spectral filter | CWL = 354.71 nm, FWHM = 0.22 nm, OOB: OD >4.0 |



| **Raman** | Optics | Ø150 mm F/4 Newton reflector |
|---|---|---|
| **receiver** | Spatial filter | Ø1.67 mrad FOV |
| | Fiber optics | Ø1 mm, 2-m long, OH-rich multimode fiber |
| | VR spectral filters | 365 nm longpass (OD >2) + 395 nm (OD >2) beamsplitter + Materion$^{TM}$ interference filters: |
| | | $N_2$: CWL = 386.76 nm, FWHM = 0.27 nm, OOB: OD >4.0 |
| | | $H_2O$: CWL = 407.59 nm, FWHM = 0.34 nm, OOB: OD >4.0 |
| | RR spectral filters | 365 nm shortpass (OD >2)  + |
| | | CWL = 355 nm, FWHM = 10 nm, flat-top, OOB: OD >6.0 + |
| | | 50:50 non-polarizing beamsplitter  + Materion$^{TM}$ interference filters: |
| | | RR1: CWL = 354.09 nm, FWHM = 0.24 nm, OD >6.0 at 354.7 nm |
| | | RR2: CWL = 353.22 nm, FWHM = 0.54 nm |
| **Detection** | Photodetectors | Hamamatsu H10721-210 photomultiplier tubes (PMT) with >0.13 A/W cathode sensitivity |
| | Amplification | Up to $2 \cdot 10^6$. Elastic & RR: fixed, VR: sky-background piloted |
| | Acquisition | 3x NI$^{TM}$ PXI-5124 two-channel digitizers |
| | | Sampling frequency: 200 MHz, 12-bit, Q-switch-trigggered |
| | Recording | 1000 shots ($\Delta t_0$ = 1 min), 200 MHz ($\Delta z_0$ = 0.75 m) |
| | | Analog + photon-counting |

## 3.1  Emitter


The emitter is a commercial Lumibird/Quantel "Q-Smart 450" Nd:YAG pulsed laser, stabilized
by injecting the output of a single longitudinal mode fiber laser emitting at 1064.175 nm into
the main cavity ("SLM" option), and frequency-tripled to emit at wavelength
$\lambda_{laser}$ = 354.725 nm (in vacuum). The nominal pulse energy for the Q-Smart 450 with SLM is
100 mJ at 355 nm, with a Pulse Repetition Frequency (PRF) of 20 Hz. These values set WALI
near the eye safety limit for pulsed energy, making the system eyesafe at the output of a 2-meter
funnel due to built-in leaks at 532 nm.
A critical issue to be cleared before using the Q-Smart 450 SLM in WALI was the spectral
purity and stability of the laser, in terms of linewidth and wavelength drift. The laser seeder at
1064.175 nm is specified with a 50 MHz (0.062 pm at 355nm) stability at fixed temperature,
and 37 MHz °C$^{-1}$ (0.046 pm °C$^{-1}$ at 355 nm) temperature drift.





Nevertheless, the stability of the Q-Smart emission at 354.725 nm has been verified with a
dedicated optical setup, sending the output of a Michelson interferometer with optical path
differences (OPD) between 0 and 100 mm on a UV-sensitive CCD camera. By extracting the
contrast and phase variations of the fringes at large OPDs from the videos, we were able to
ascertain:
•   the laser linewidth, without seeder, to be 24±2 pm (versus 26.5 pm datasheet value), and
with seeder, to be small compared to 1 pm (versus 0.2 pm datasheet value),
•   the wavelength drift, without seeder, to be below 8 pm over 10 minutes, and with seeder,
to be below 0.2 pm RMS (root mean square fluctuations) over 5 minutes. We consider
the remaining fluctuations to be due to the ~0.05 pm °C$^{-1}$ temperature-linked drift of
the seeder, which is not temperature-controlled.
Given the requirements derived in Section 2.3, this makes the seeded Q-Smart laser
theoretically suitable for RR measurements of temperature.
**3.2   Raman receiver**
In this sub-section we discuss the possible impact on the VR/RR ratios of the fibered reflector
(beam scrambling and fiber optics fluorescence), of Raman filters characteristics, and of the
polychromator design and alignment. As far as we know, this type of comprehensive study does
not exist in the literature for Raman lidars.
3.2.1 Fibered reflector telescope and scrambling of the lidar field-of-view
The elastic and Raman receivers are both 150 mm in diameter. The focal length of the refractor
(elastic channels) is ~300 mm, which with a 200x600 µm field stop achieves full overlap at
~150-200m. However, the focal length of the reflector (Raman channels) is 600 mm (parabolic
mirror with aperture F/4); this implies using a multimode fiber optics about 1 mm in diameter
as the field stop to allow similar results in terms of field-of-view and overlap. The chosen fiber
optics is an OH-rich UV fused silica fiber, 2 m in length and 1000 µm in core diameter, with
numerical aperture 0.22 (Avantes FC-UV1000-2).
Coupling the reflector output into a multimode fiber (e.g. Chourdakis et al., 2002) allows: i) to
minimize occultation of the primary mirror (here only 12 mm in diameter), ii) to deport the
Raman channel separation away from the telescope, making it a separately tunable optical
system, minimizing the overall lidar size and making light or temperature confinement easier,
iii) in theory, to scramble the fiber output illumination versus the lidar field angle, therefore



minimizing the range-dependence of AOIs on the IFs discussed in Section 2.3, and flattening
overlap ratios after the geometrical full-overlap distance.
The scrambling of the lidar field-of-view, via the multiple internal reflections in the fiber, has
been experimentally tested by imaging the output of the fiber, with a varying point-like input.
The results are shown in Figure 3. Note that the radial coordinate of the output point relative to
the center of the fiber corresponds to a given AOI on a well-aligned IF in the following
polychromator, after a $f' = 50$ mm doublet lens.

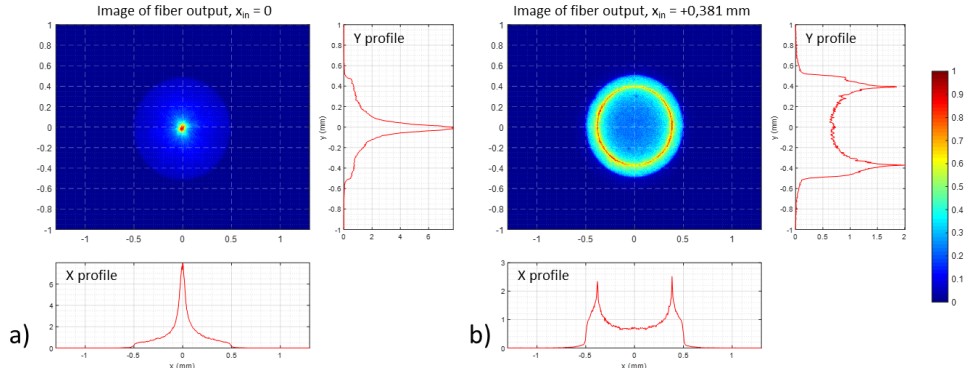


Figure 3. Images of the output facet of the 1 mm diameter multimode fiber optics for a) centered
and b) decentered (at $x_{in} = 0.38$ mm horizontal offset from the center of the core) input point of
a 20-mm beam focused on the input facet of the fiber, and energy density profiles along the x
and y axes.
It appears on Figure 3 b) that the input energy is mostly redistributed tangentially (i.e. along the
angular polar coordinate, as opposed to radially) by its passage through the fiber. The radial
dispersion remains small, and the mean output radius is approximately equal to the input radial
coordinate. Manually applying curvature to the fiber, as suggested by so-called "mode
scrambling" devices, did not make the energy distribution more uniform so much as creating
unwanted losses (effect not shown). Even for a centered input, the energy radial distribution –
i.e. the percentage of the total output in a given radial bin, that will therefore impact a well-
aligned filter at the same AOI – is uniform. We conclude that while minimizing effects of filter
misalignment, the use of fiber optics does not substantially make the angle of incidence on the
interference filters independent from the image position in the focal plane of the telescope, in
contrast to what could be expected. Range-dependent biases will not be strongly mitigated.
### 3.2.2  Fiber optics fluorescence
It has been shown by Sherlock et al. (1999) and discussed by Whiteman et al. (2012) that fiber
optics fluorescence could be an obstacle to water vapor measurements, because elastic
scattering at 532 nm was inducing fluorescence in an OH-poor fiber at a non-negligible level
compared to the atmospheric Raman scattering. It was solved by using an OH-rich fiber, but it
was predicted in the latter work that the effect could be larger at 355 nm.
We have characterized this effect in the WALI fiber optics, using a narrowband CW laser
excitation centered at 355 nm. The output of the fiber was analyzed by a Fourier transform
spectrometer (Thorlabs OSA201C spectrum analyzer), behind a longpass dichroic plate cutting
the direct LED emission, and the same collimating achromat as in the polychromator. The
resulting spectrum is shown in Figure 4.

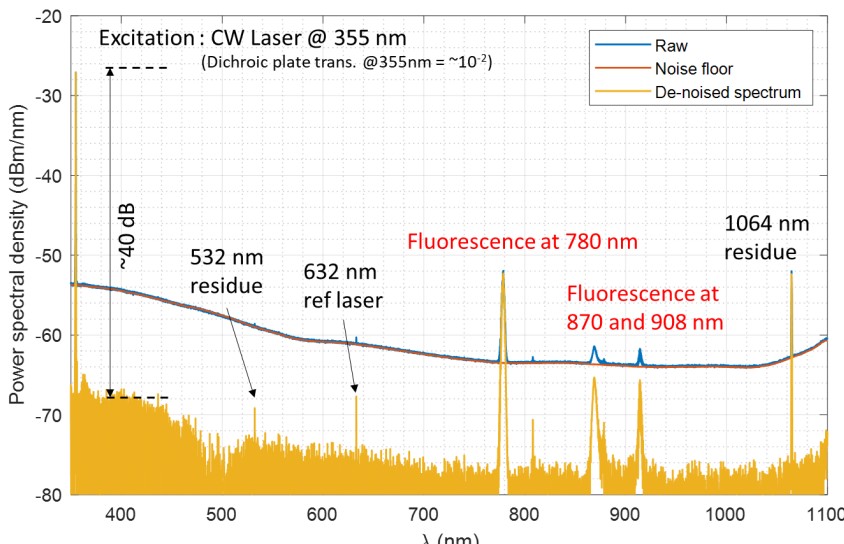


Figure 4. 1000-µm diameter, 2-meter long fiber fluorescence measurement with 355 nm laser
illumination.
We plot both the raw spectrum and the Fourier transform spectrometer noise floor after 1000
profile integrations, to highlight the very weak features observed at 780 to 910 nm, and the high
associated uncertainty. Due to the noise level, and given the dichroic plate residual
transmittance of the laser wavelength, we can only ascertain that the fluorescence power
spectral density (PSD) around 400 nm is lower than $10^{-6}$ times the peak laser PSD, although no
feature can be detected in this spectral domain. Note that fluorescence between 400 and 500 nm
was indeed observed using a broadband excitation from a fibered LED at 340 nm (not shown).
Nevertheless, the amount of rejection observed for a 355 nm excitation is sufficient to exclude
an adverse impact of the OH-rich fiber optics for Raman lidar measurements.
**3.3   Raman channels**
3.3.1 Polychromator configuration
The RR+VR polychromator configuration used in WALI is presented in Figure 5. Dichroic and
non-polarizing beamsplitters are used to separate the channels. In contrast to the design of
Hammann et al. (2015) which optimizes throughput and laser-line rejection on the RR channels,
we chose to implement a splitter-based configuration, favouring a compact system (25x25 cm,
easier to confine) and normal incidence on the filter, at the expense of SNR. Indeed, designing
the filters for a correct CWL at 5° incidence (as in the cited work) instead of 0° dramatically
narrows the filter angular acceptance, as can be deduced by deriving Eq. (15) as a function of
incidence $\theta'$. In the WALI polychromator, the output from the fiber is collimated by a near-UV
achromat with 50 mm focal length, resulting in a 22 mm diameter beam. Dichroic beamsplitters
with adequate cut-on wavelengths are used to separate channels. On each separated channel, an
aspheric lense condenses light on the PMT surface, located 4 mm before the focal plane. A cage
system assembly holds all parts with great stability, however beamsplitters are not always
perfectly aligned at 45° in the stock cage cubes. That is why all filter, lens and PMT sub-
assemblies are mounted on tiltable mounts to allow precise alignment at normal incidence.

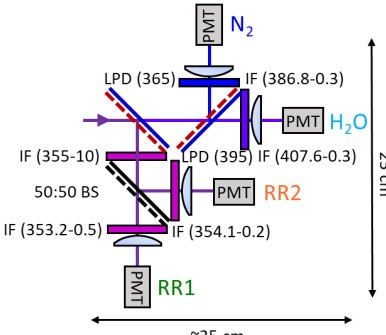


Figure 5. Compact rotational & vibrational Raman separation configuration used in WALI. IF:
interference filter (with CWL - FWHM given in nm), OD: optical density, BS: beam splitter,
LPD: long-pass dichroic beamsplitter (with cut-off wavelength given in nm), PMT: photo-
multiplier tube. This polychromator is thermally regulated in a dedicated light-tight enclosure.



### 3.3.2 Filters qualification

All interference filters were custom-made by Materion, including the RR filters on specifications graciously shared by the team of A. Behrendt (following Hamann et al., 2015). They were characterized on the Fourier transform spectrometer (described in section 3.2.2) prior to mounting, using fibered LEDs peaking at 340, 385 and 405 nm as the light source; the beam was collimated by the same near-UV achromat with 50 mm focal length. We give the measurements results for the RR filters in Figure 6 and Table 3.

The effective index and angular acceptance of the filters (arbitrarily chosen for a 10% loss at the CWL) were assessed by tilting the filters of a known angle. A critical parameter, the transmittance of both filters at the laser line $\lambda_{laser}$ in operational conditions was assessed on the lidar itself, by measuring the energy of an echo on a hard target located at 200 m, and switching between an elastic IF of known transmittance with a known strong optical density and the RR IF in question. The excellent extinction in the RR1 filter guarantees a minimal effect of elastic signal leak in temperature retrievals, but it was nevertheless subtracted as in Eq. (3). Note that no significant echo was detected on the $H_2O$-Raman channel, indicating extinction better than a few $10^{-9}$, thanks to the two dichroic plates.

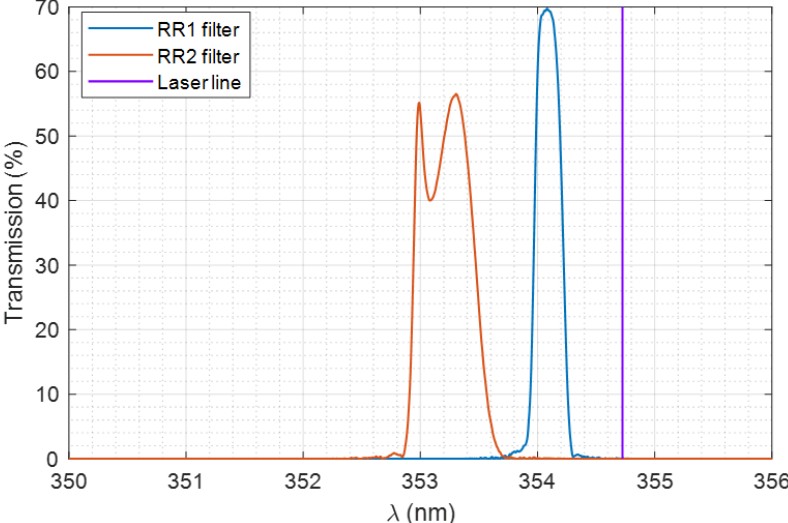

Figure 6. RR filters spectral transmittance measured on optical spectrum analyzer with illumination by a 340 nm LED: RR1 (low-J) and RR2 (high-J) filter at 0° incidence.

Table 3. Measured RR IF characteristics. All CWL values are given in vacuum.



| | RR1 filter | RR2 filter | Uncertainty |
|---|---|---|---|
| CWL | 354.09 nm | 353.22 nm | 0.01 nm |
| FWHM | 0.24 nm | 0.54 nm | 0.01 nm |
| $n_{eff}$ | 1.62 | 2.03 | 0.05 |
| Max transmittance | 69% | 51% | 5% |
| Laser line transmittance | $2.7 \ 10^{-8}$ | $2.9 \ 10^{-7}$ | 10% relative |
| Angular acceptance (AOI for 10% loss at CWL) | 1.5° | 2.5° | 0.2° |
| CWL shift at max field angle (i.e. edge of fiber, AOI = 0.59°) | -9.8 pm | -2.5 pm | 0.3 pm |

### 3.3.3 Polychromator alignment and qualification

Due to the filter CWL shift evolving as the square of the AOI in Eq. (15), it is essential to minimize range-dependent biases by aligning the filters at a precisely normal incidence from the input beam. However off-the-shelf beam splitter plate holders are found to be misaligned by up to 1° from an ideal 45° incidence. All PMTs are mounted jointly with their own IF and lens into a tiltable mount to correct for this (represented on Figure 7).

The alignment of these mounts is performed in the lab by conjugating an input multimode fiber of 600 µm diameter replacing the lidar input, into a target fiber 200 µm in diameter at the focus of the PMT lens, through the polychromator. Fibered LEDs are used for illumination like in section 3.3.2. All the channels are sequentially addressed in this manner. By obtaining a maximal energy and a radially uniform profile at the output of the target fiber, one can ensure alignment with a precision of 0.1 to 0.3°.

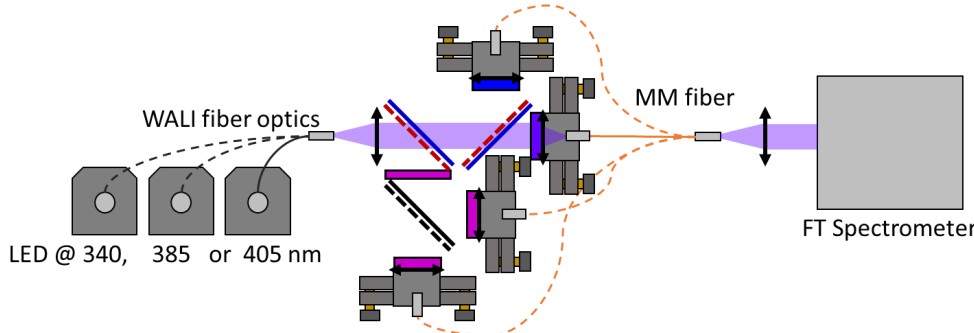





Figure 7. Method for polychromator alignment validation. Light from LEDs is input in the
WALI fiber optics, passes through the polychromator, and into a multi-mode fiber (MM fiber,
Ø600 µm) analysed by a Thorlabs OSA201C Fourier transform spectrometer. Channel central
wavelengths are expected not deviate from those of the filter measured independently at normal
incidence, to validate alignment.
To verify the result, the spectral transmittance of the polychromator channels themselves are
characterized by the Fourier transform spectrometer, as shown on Figure 7. By illuminating the
channel with a LED coupled in the actual lidar fiber, we ensure that the polychromator is studied
in operational conditions. The CWL of each channel is expected not deviate by more than 20
pm (twice the empirical accuracy) from the CWL measured on the individual filter at normal
incidence, to validate the alignment. The polychromator aligned using the procedure proposed
above passes this test.
**3.4   Detectors**
Hamamatsu 10721P-210 PMTs, with >0.13 A W$^{-1}$ cathode sensitivity at 400 nm, and up to
~2 10$^6$ controllable internal gain, are used to transform the optical flux into an electric current,
directly digitized at 200 MHz (0.75 m sampling along the line of sight) by three NI PXI-5124
two-channel digitizers with 50 Ω load. The acquisition software, custom-made with Labview,
conducts analog and photon-counting (thresholding at ~3 standard deviations of the noise)
accumulations in parallel during 1000 shots (50 seconds), every minute, which are then pre-
processed and recorded (~10 seconds down time). Every ~8 minutes, baselines are recorded
with PMT gains set at zero. The next sub-sections describe critical points of the detectors
affecting the RR and VR channel ratios.
3.4.1  PMT response variability
As explained in Section 2.3, the non-uniformity of the PMT response can affect the ratios of
Raman channels as a function of range. We tested the sensitivity profiles of WALI's $H_2O$-
Raman PMT to continuous laser illumination at 405 nm wavelength, first using a 1-mm
diameter collimated beam, as a function of both point and angle of incidence. A cumulated ~6.0
optical density was used to avoid saturation of the PMT.
As shown on Figure 8 a), a strong variation of sensitivity by a factor of almost 2 is found on the
PMT surface, much larger than specified. The relative sensitivity is lowest near the center of
the PMT and highest on the sides, on a diameter of 4 mm approximately equal to the spot size
in the lidar. Indeed the PMT surface is 4 mm before the focal plane of the 0.5 NA condensing





aspheric lens. This is consistent with the results of Simeonov et al. (1999) on an older generation
of detectors, excluding a suspected hole-burning phenomenon over the lifetime of our PMT.
On the vertical axis, we also note the effect of the gridded cathode. Note that sensitivity does
not vary by than a few percent as a function of angle of incidence (not shown).

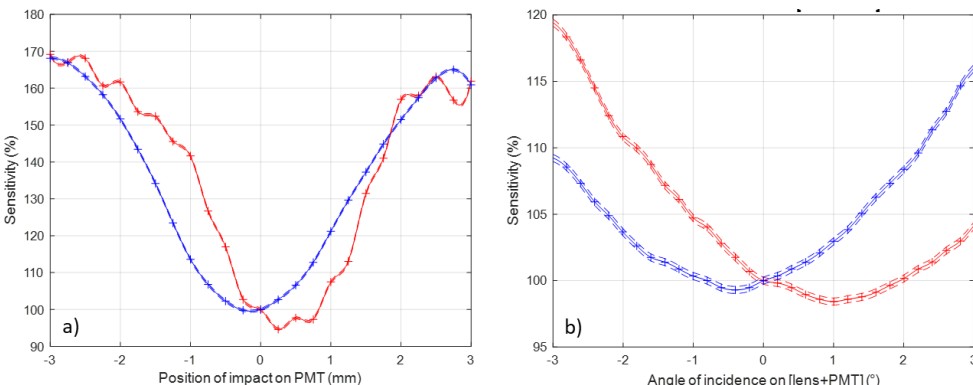


Figure 8. Study of the non-uniformity of the PMT response: a) as a function of point of impact
on the active area along the horizontal (blue) and vertical (red), with a 2 mm collimated beam
from a 405 nm laser, b) as a function of angle of incidence on the lens and PMT assembly
similar to the ones used in the WALI polychromator, with a 21 mm collimated beam from the
same laser. Dashed lines represent uncertainty calculated over multiple measurements.
We then put the condensing lens used in the polychromator in front of the PMT, and studied its
response as a function of AOI on the lens+PMT assembly, which is shown in Figure 8 b). The
input beam was the nominal size in the polychromator ie. ~22 mm in diameter. We find that the
curve corresponds well to the measured sensitivity profile, smoothed by its convolution by the
spot on the PMT. The problem is that at normal incidence, the derivative of sensitivity with
incidence is 2-5% per degree. In the future, the condensing lenses will be replaced with afocal
beam reducers to reduce this dependency.
3.4.2  Baseline and EM parasites correction
The baseline induced by the detection chain is found to vary between channels and in time. It
is also subject to electro-magnetic (EM) interference causing parasitic signals of both high
frequency, mostly due to the flashlamp high peak current radiating over the system, and low
frequency, probably due to other neighboring electronics. For this reason, the channel baselines
are evaluated regularly (by averaging 1000 shots with PMT gain set to zero, every 8 minutes),
smoothed and corrected ($L_j$ in Eq. (2)). However, for the Raman channels (H$_2$O and RR2
specifically), the weakness of the signals requires a specific care of EM compatibility, as



repeating parasitic spikes were found to jam the channels (especially photon counting) starting
at altitude 6-7 km.
Figure 9 a) shows an example of perturbed baseline. Trial and error established that common
methods to avoid ground loops were not all efficient: star grounding of the various cables
worsened the problem, whereas physically separating coaxial signal cables from direct current
power supply and control voltage cables, and grounding all connectors and opto-mechanics
again on the breadboard side, mitigated it, reaching the baseline plotted in Figure 9 b).

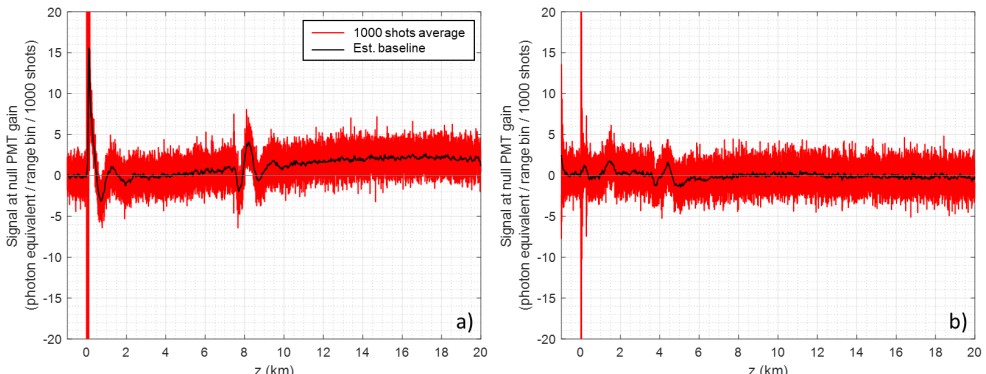


Figure 9. Analog detection baseline measurements (red) over 1000 laser shots with PMT gains
set to zero, expressed in photon counts equivalent on the RR2 channel: a) in an unfavorable
case (no mitigation), showing both baseline fluctuations over time (20 km ~ 133 µs) and strong
electro-magnetic parasites at large distance; b) on the WALI system, after mitigation. The final
estimated baseline ($\hat{L}_j(z)$ in Eq. (3)) obtained after smoothing, which is subtracted to all
recorded profiles, is in black.

### 3.4.3  PMT gain adaptation

On each channel, PMT internal amplification gain $G$ (using photoelectron multiplication) is a
definite function of its control voltage $U$. The variation of $G$ by ~2 orders of magnitude allows
for the optimization of the dynamic range. This helps deal with the different Raman cross-
sections in each filter, with variations of atmospheric transmittance, and especially with sky
background levels during daytime. The gain is pushed at its maximum possible value still
satisfying two conditions: i) the signal voltage maximum does not exceed the range of the
digitizer, ii) the sky background signal does not exceed the maximum output current of the PMT
that guarantees linearity (100 µA, ie. $<S_{raw}> < 5$ mV). This is indispensable for day-round





measurements of WVMR, otherwise the channels would be saturated during daytime (Chazette
et al., 2014b), or suboptimal in SNR during nighttime.

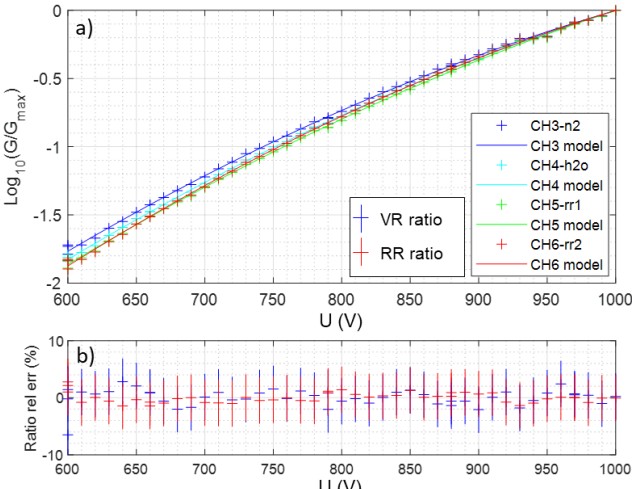

Figure 10. Calibration of PMT gain $G$ versus control voltage $U$: a) log-gain measurements and
second-degree polynomial model for all Raman channels, and b) relative gain ratio error
between model and measurements for vibrational and rotational Raman channel ratios.
However, PMT gain adaptation leas to biases on the Raman channel ratios if the gain versus
control voltage characteristics are not known with a better precision than the requirements
stated in Table 1 (2% on VR channels, 0.4% on RR channels). In Figure 10 a), we show the
experimental calibration of $G$ versus $U$ as well as second-degree polynomial fits for each
channel. The relative error on the VR and RR channel gain ratios approximated by these models
is plotted on Figure 10 b), with the measurement uncertainty. This uncertainty is mostly due to
variations of atmospheric parameters and laser energy during calibration. Since all relative
errors are well centered, we compute that the possible error for the gain ratio with these models
is ~1.3 %. This is compatible with WVMR measurements but not with temperature
measurements. Therefore, the PMT gain should only be adapted on the VR channels, and the
RR channels should be kept at a fix value of gain.
### 3.4.4 Merging analog and photon-counting signals
Both analog and photon-counting raw signals are recorded. The analog signal has lesser SNR
at high altitude during nighttime, whereas the photon-counting signal is saturated at low altitude
and by daylight; by merging them correctly, an optimal SNR can be obtained (Newsom et al.,



2009). For signal processing, the photon-counting raw signals are first desaturated (details in
Chazette et al., 2014b). Merging is performed during nighttime on the pre-processed signals
defined in Eq. (3). After calculating a photon to Volts conversion constant at an altitude where
photon-counting is not saturated, the converted photon-counting profile replaces the analog
profile after a predefined altitude depending on signal strength (from 1 km for the $H_2O$ VR
channel, up to 4 km for the elastic channel).
We wish to emphasize here that baselines $L_j$ and background signals $B_j$ in Eq. (3) must be
estimated separately for the analog and photon-counting recorded profiles (which have no
baseline, and a smaller but non-zero background value due to the suppression of electronic
noise). Otherwise, the merged signal will show discontinuities at the cut-off altitude, and biases
at high altitude at dusk and dawn. Their impacts are typically much larger than the requirements
of Section 2.2.

## 4  Qualification on the atmosphere

In this section, we qualify the WALI system starting with the measurement of its overlap factor
ratios, followed by its calibration and comparisons with radiosoundings. Remaining biases are
highlighted and corrected, and experimental measurement errors are evaluated.

### 4.1  Experimental set-up and strategy

We put the lidar into operation in our laboratory near Saclay (48°42'42"N 2°08'54"E) over a
period of two weeks in May 2020. It was placed on a rotating platform below a trapdoor
equipped with silica windows for zenith shots, and in front of a window at a height of about 9
m above the ground level (agl) for horizontal shots. During the latter, the lidar aimed North <5°
above the horizon (beam elevation <80 m per km of range). In that direction, land use is fields
up to 800 m range, buildings and trees between 800 and 2 km range, and fields again up to 5.5
km range.
To calibrate and qualify the lidar measurements, we use radiosoundings launched two to three
times daily from the operational Météo-France station located in Trappes (48°46'27"N
2°00'35"E, 12.3 km WNW from the lidar near Saclay, approximately upstream in the prevailing
winds).



### 4.2 Measurement of overlap ratios with horizontal shots


The overlap factors and their ratios were estimated on signals averaged over 3 hours after sunset
on December 19th, 2019, with a rather lukewarm, unturbulent but hazy atmosphere (aerosol
extinction coefficient $0.32\,\text{km}^{-1}$ at 355 nm with Angström exponent ~1.5, 11°C ground
temperature, and WVMR at ground level around $6.5\,\text{g}\,\text{kg}^{-1}$). With a planetary boundary layer
(PBL) height of ~900 to 1000 m, and slow gradients of temperature (−1 to −4°C $\text{km}^{-1}$) and
WVMR (−0.8 to −1.2 $\text{g}\,\text{kg}^{-1}\,\text{km}^{-1}$) in that PBL (as measured by radiosoundings launched from
Trappes at ~12:00UTC and 0:00UTC, presented in the next subsection), conditions were
excellent for a homogeneous atmosphere within the first 5 km at least.
The estimated overlap factors of the different channel, with atmospheric extinction fitted
between 800 and 2000 m, are shown in Figure 11 a). Full geometrical overlap is obtained as
expected between 150 and 200 m, but the curves differ by several percent between the Raman
channels. Atmospheric extinction drifts from the estimated value after 2 km.
The estimated ratios of overlap factors $OR_R$ and $OR_Q$ are plotted in Figure 11 b) and c), at 7.5 m
resolution (thin line) and after smoothing (thick line, final correction used hereafter). Peak
divergence is 5 to 7 %, at ~150 m. Convergence within 1% happens at ~400 m, but oscillations
of lower amplitude persist until ~3 km. We note that for $OR_Q$, deviations do not exceed the
±0.7% required to maintain bias below 1°C. They are nevertheless corrected.

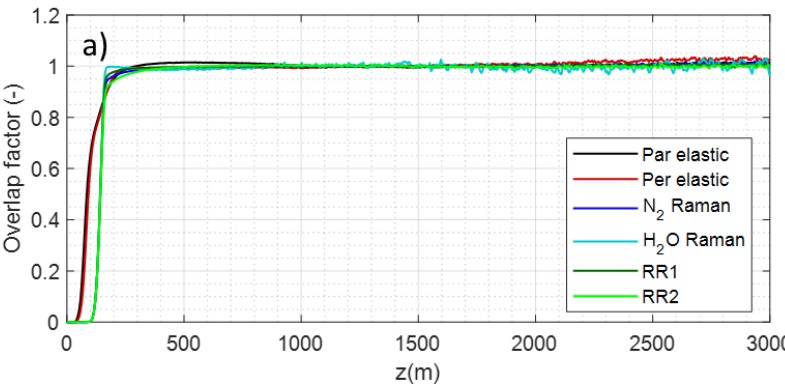




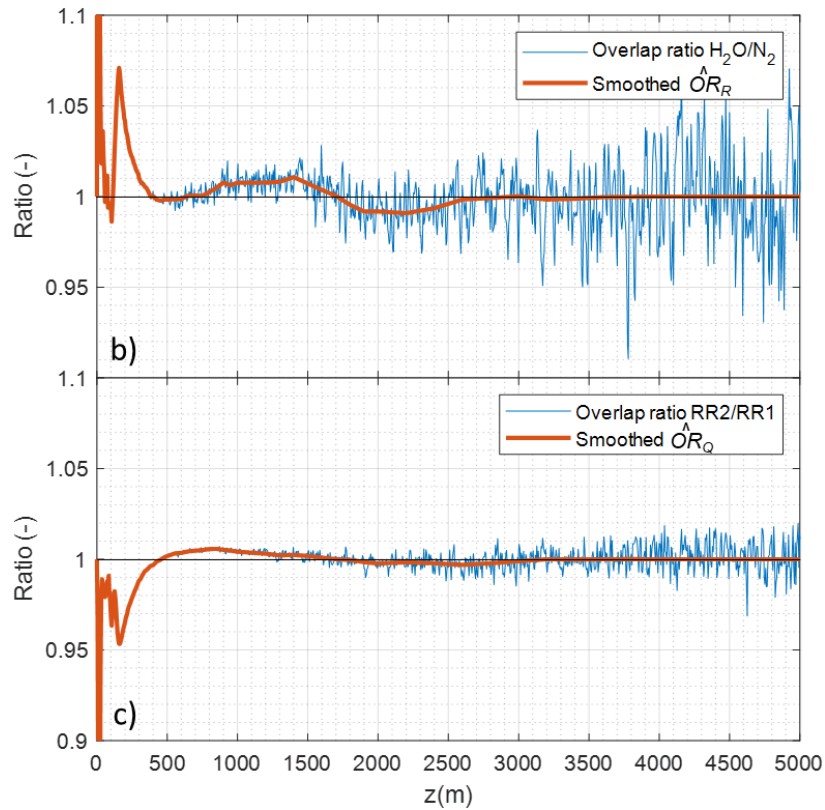


Figure 11. a) Overlap factors measured over 3 hours of nighttime measurements with a
horizontal line-of-sight on Dec. 19, 2019. Estimated overlap ratios between VR (b) and RR (c)
channels: native resolution (thin blue line), final estimate after smoothing (thick red line).
**4.3   Comparison to radiosoundings and calibration, estimation of residual error**
12 nighttime and 24 daytime radiosoundings were launched from Trappes between May 20th
and June 2nd, 2020. Lidar profiles are averaged from 0 to 40 minutes after the radiosounding
launch time. The range averaging is progressive and defined to keep the night time temperature
error below 1.5°C: range bins are 15 m long below 100 m agl, growing to 360 m above 8 km
agl.
In order to debias WVMR and temperature measurements from residual errors on $OR_R$ and
$OR_Q$, we perform a three-step calibration:





• First step: we exclude the first 1500 m agl of the profiles when fitting $r_{H2O}$ *in-situ* vs $R'$

and $Q'$ vs $T$ *in-situ* to estimate $K$ and $f$ respectively. This initial calibration is shown in

Figure 12 a) & d).

• Second step: using these first estimates, we then plot the ratios between the lidar

observables $R'$ & $Q'$ and the expected observables deduced from the in-situ

measurements and these initial calibration parameters. This provides an estimate of the

remaining biases on $OR_R$ and $OR_Q$, which we find to be up to ~4% and ~1.8%

respectively. This represents a small correction to the overlap ratios estimated while

shooting horizontally, but remains larger than the requirements of precision specified in

Table 1. The modeled corrections of $OR_R$ and $OR_Q$ are plotted in red in Figure 12 b) &

e).

• Third step: we apply the previous estimates of $OR_R$ and $OR_Q$ and we perform a new

calibration using all the data (down to 200 m agl), yielding more precise estimates of

calibration constants, as shown in Figure 12 c) & f).

In the three steps, data with SNR lower than 10 for $R'$ and 30 for $Q'$ are rejected so as to
limit the impact of noise present at higher altitudes.

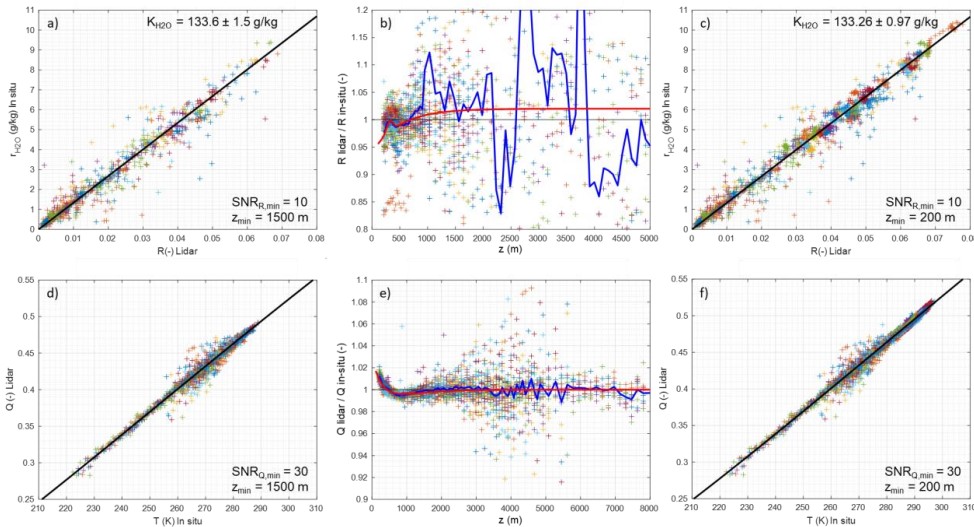

Figure 12. Results of calibration on 12 nighttime and 24 daytime radiosoundings launched from
Trappes between May 20th and June 2nd, 2020 for WVMR (upper row) and temperature (lower
row), in three steps: calibration on measurements above 1500 m (a/d) with samples as crosses
(one color per radiosonde) and calibration curve in black; residual overlap ratio estimation (b/e)



with samples as crosses, mean ratio in blue and model in red; calibration on all results (c/f).
Daytime samples are limited to SNRs above 10 for *R'* (WVMR) and 30 for *Q'* (temperature).
The reliability of this calibration along time has been tested by comparing to the same exercise
performed two months later at the end of July 2020. After calibration in the same conditions
than in May, we found $K$ decreased by ~7.3%, and the temperature associated to a given value
of *Q'* to be ~2.1°C higher. However, $OR_R$ and $OR_Q$ were still accurate within the reachable
precision, ie. ~0.2%. It was later proven that a malfunction of the laser seeder was responsible
for a slow drift of the emitted wavelength. Thus, although a regular verification of the
calibration is necessary, the measurement of the overlap ratios is reliable.

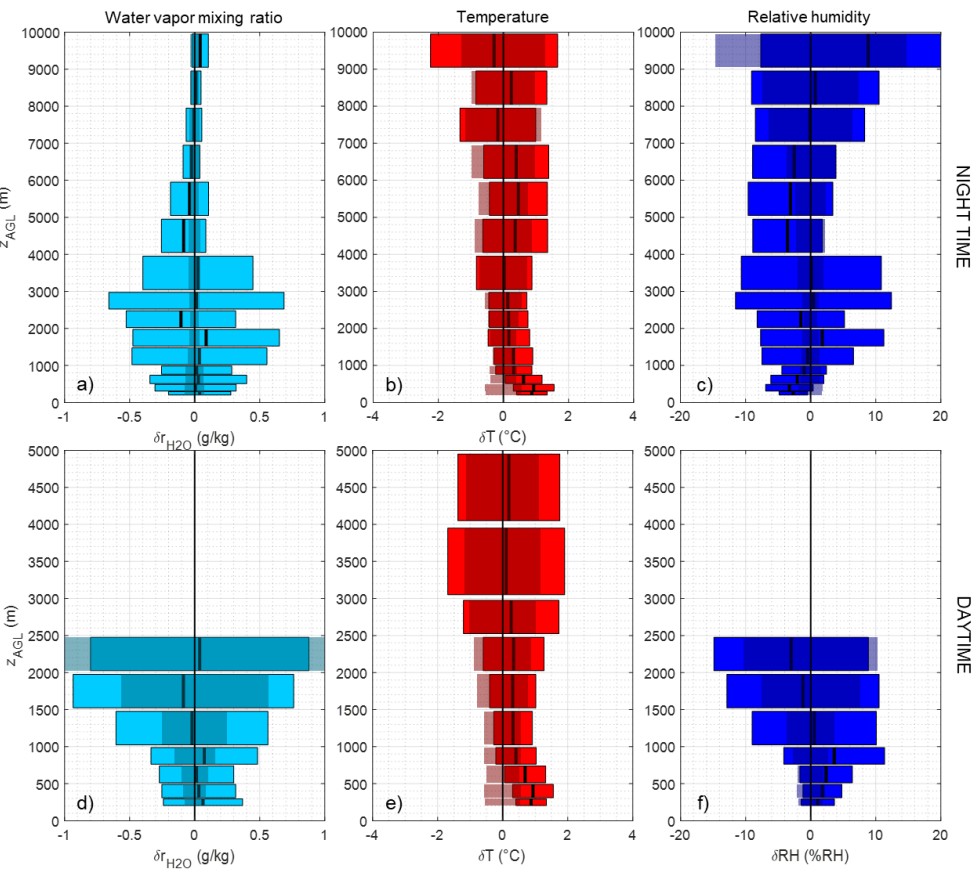


Figure 13. Residual deviations between lidar and Trappes radiosoundings in terms of WVMR,
temperature and relative humidity, for night time (a/b/c) and daytime (d/e/f), with mean
deviation (thick lines), and RMS error (colored rectangles). The error corresponding to noise





levels on the lidar signal is shown as darker rectangles. Cloudy profiles have been discarded.
Daytime measurements are limited to SNRs above 5 for $R$ (WVMR) and 20 for $Q$ (temperature).
In Figure 13, we examine the residual deviations between the lidar and the same series of
radiosoundings used for the calibration. RH has been derived using Eq. (14) from lidar-
estimated WVMR and temperature, and the pressure profile given by radiosoundings. For each
parameter $r_{H2O}$, $T$ and $RH$, we plot for daytime and night time profiles the mean and RMS
deviations averaged over large range bins as colored bars, as well as the propagated signal error
as darker shaded areas. This allows to compare the observed random error to what could be
expected from the level of noise on the lidar measurements. Note that only profiles with good
SNR unperturbed by clouds have been selected for this comparison.
On WVMR, the results show little bias, and RMS deviation is dominated by atmospheric
variability by night at low altitude and by lidar noise otherwise. On temperature, most of the
RMS deviation is explained by noise; a ~1°C significant bias is still seen below 800 m. This
seems to be due to an underestimated correction of $OR_Q$. On Figure 13 c) and f) are plotted the
consequences of this bias on relative humidity $RH$ to be around 2 to 4%RH, but also the
resulting error to be expected. We see that with the defined averaging, random error is around
2% RH up to 5 km agl during nighttime and 1 km agl during daytime, growing fast above.
Table 4. Statistics of observed differences for $r_{H2O}$, $T$, and $RH$: experimental Mean Differences
(MD), Root-Mean Square Differences (RMSD), averaged over two different range bins, in the
low troposphere (1-2 km) and the free troposphere (5-6 km). Comparison to the "natural"
atmospheric variability between the lidar and RS sites as modelled by the ECMWF/IFS ERA5
reanalyses (difference over considered period between grid points nearest to each of the two
sites), and to the theoretical root-mean-square error (RMSE) derived from the variance of the
RR signals.

|  | Range | Range resolution $\Delta z$ | Model atmos. MD | Experimental MD (night/day) | Model atmos. RMSD | Theo. RMSE (night/day) | Experimental RMSD (night/day) |
|---|---|---|---|---|---|---|---|
| **WVMR (g/kg)** | 1-2 km | 84 m | -0.03 | +0.06/-0.05 | 0.41 | 0.03/0.4 | 0.54/0.65 |
|  | 5-6 km | 168 m | $<10^{-2}$ | -0.07 | 0.11 | 0.04 | 0.15 |
| **Temperature (°C)** | 1-2 km | 84 m | +0.15 | +0.25/+0.3 | 0.33 | 0.4/0.7 | 0.6/0.7 |
|  | 5-6 km | 168 m | +0.05 | +0.4 | 0.28 | 0.75 | 0.95 |





| Relative humidity (%RH) | 1-2 km | 84 m | -0.23 | +0.8/-0.5 | 6.37 | 1.7/5.5 | 6.5/10 |
|---|---|---|---|---|---|---|---|
| | 5-6 km | 168 m | -0.70 | -3.3 | 7.52 | 2.2 | 7 |


To support the above interpretation, in Table 4 we compare the experimental mean difference
and RMS difference plotted on Figure 13, averaged over two altitude ranges (low troposphere,
LT, 1 to 2 km, and free troposphere, FT, 5 to 6 km), to i) the natural variability of the atmosphere
between the radiosondes at Trappes and the lidar at LSCE, as modelled by ERA5 reanalyses of
the ECMWF/IFS weather model, ii) the expected random error given the noise level on the RR
signals. Nighttime and daytime values are indicated in the LT, only nighttime values in the FT.
We see that the experimentally observed values of RMSD are rather consistent with the
quadratic sum of the RMS variability of the atmospheric variables between Trappes and LSCE,
and of the noise-induced RMS error. The excess random difference is thus well explained by
the distance. There is still a discrepancy with the mean difference of temperature however; it is
not explained by the modelled differences of temperature at the locations of the two soundings.

## 5 Conclusion

During the qualification of the rotational Raman channels for the WALI lidar of LSCE, with
the aim of providing profiles of relative humidity, we encountered important sources of bias
that are seldom described in the now abundant literature involving such systems. We
highlighted the predominant effects of the dependency of filter transmittance and detector
sensitivity upon angle of incidence and point of impact, respectively. Because the latter
parameters are directly proportional to field angle, they cause range-dependent biases on the
RR/VR signal ratios that are several times greater than the required accuracy of lidars for
temperature measurements (only 0.79% for 1°C here), less so for water vapor measurements.
We established that this effect cannot be suppressed by using fiber optics between the receiver
and polychromator, because scrambling of the lidar field of view does not happen radially in
the fiber. Mitigation efforts impose the careful alignment of each filter at normal incidence to
the input beam, and the verification of the spectral transmittance of each channel on a
spectrometer. The thermal stability of the polychromator is also of prime importance. Other
significant bias sources include electro-magnetic perturbations of signal baselines and PMT
gain variation, which must be mitigated. The impact of fiber optics fluorescence, and of the
measured laser linewidth or short-term wavelength drift were shown to be negligible in the
WALI system.
After a measurement of RR/VR channel ratios during horizontal shots, which showed the
significant impact of the above phenomena (up to 5% bias on ratios below 300m, ~1% higher),
we calibrated and de-biased the WALI measurements using radiosondes launched from the
nearby Trappes station of Météo-France. Between the de-clouded lidar measurements and the
radiosonde profiles, the remaining mean differences are small (below 0.1 g/kg on water vapor,
1°C on temperature) and RMS differences are consistent with the expected error from lidar
noise, calibration uncertainty, and horizontal inhomogeneities of the fields between the lidar
and radiosondes. On relative humidity we thus reach a goal of ~10%RH random error and
5%RH systematic error up to 9 km by night and 1.5 km by day, with 40 min time integration
and progressive vertical integration of 15 to 360 m at 10 km. The systematic error on RH is
dominated by bias on temperature, whereas the random error is dominated by noise on water
vapor measurements.
Thus exhaustively qualified, the WALI system may be applied in the near future to exercises
assimilating thermodynamic profiles in weather models, as is expected within the WaLiNeAs
(Water vapor Lidar Network Assimilation experiment) project (Flamant et al., 2021).

## Acknowledgements

The authors thank Andreas Behrendt and Diego Lange for sharing specifications and for useful
discussion. Radiosoundings from the Tessereinc de Bort station (Trappes) were obtained at
https://donneespubliques.meteofrance.fr/?fond=produit&id_produit=97&id_rubrique=33,
courtesy of Météo France. ERA5 reanalyses of the ECMWF/IFS model were obtained at
https://cds.climate.copernicus.eu/cdsapp#!/home courtesy of the Copernicus Climate Change
Service (C3S). This work was funded by the Centre National d'Etudes Spatiales (CNES), and
by the Commissariat à l'Energie Atomique et aux Energies Alternatives (CEA).

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
