# Peer review of "Mitigation of bias sources for atmospheric temperature and"

_Atmospheric Measurement Techniques, 2021_

## Referee Comment (RC2)

**General comment**

The study "Mitigation of bias sources for atmospheric temperature and humidity in the mobile Weather & Aerosol Raman Lidar (WALI)" by Julien Totems, Patrick Chazette and Alexandre Baron provides a thorough description of the the WALI system both from the point of view of the technical characterization of the lidar transciever and the performances in terms of bias and RMS. The article is well written, easy to read and exhaustive. All topics are described and supported by either previous literature or statistical studies performed by the authors. In Sect. 4.3, the comparison with the radiosounding, in addition to calibration purposes, brings useful information about the quality of the WVMR and temperature data. Accepting the 12 km distance between lidar and in-situ measurements, and then accepting a higher RMS due to slightly different atmosphere, it allows to use the bias and RMS values as solid evaluation of the WALI performance. There are only few technical comments/remarks that should be addressed before publication. I strongly recommend this manuscript to be published in AMT.

**Technical comments**

Abstract: while the abstract sets the main objectives of the study within the state of the art, it does not state any quantitative result. In this ways, the abstract fails to deliver to the reader a concise summary of the obtained results. the main results regarding calibrations and comparisons with radiosounding presented in section 4 should be reported in the abstract by stating the mean daytime and nighttime biases and RMS values.

Introduction, ln 55, 64, 71, 76: the authors mention the "sources of biases". A statistical bias is typically a systematic error, a difference between the measurement and the truth. In this sense it would be more appropriate to refer to the "sources of uncertainty" or "sources of error".

Sect.2.1, Pg 4, ln 90-91: which are the "required altitude and time"?

Sect.2.1, Pg 4, ln 101: "corrected for"

Sect.2.1, Pg 4, ln 106-109: have the authors actually did some simulation to asses the reliability of the 5%-impact of the differential extinction or the estimate by Whiteman 2003 is taken directly?

Sect.2.1, Pg 4, ln 110-111: do the authors mean that the N2 has a constant mixing ratio through troposphere and stratosphere? The statement is not formulated in a clear way.

Sect.2.2, Pg 5, ln. 126-129: the authors set the requirement for successful monitoring, verification and data assimilation into models by listing noise errors and biases. If I interpret correctly what the authors mean by bias, this should not be part of the requirement as they can be efficiently removed by the calibration process.

Sect. 2.3, pg 8, ln 187: "thus"

Sect. 2.3 pg 9, Figure 1b: the caption does not say what the green lines represent. One can imagine that is the return beam from the IF, but it is not clear.

Sect. 3.3,pg 17 ln 398: what material the cage system is made of? Is the cage subject to thermal expansion?

Sect. 3.3-3.4: the authors perform a thorough analysis of the detectors' sensitivity, calibration and responses. PMT sensitivity and gain are also analysed in detail, which

allows correcting for inequalities at the PMT output. As it is shown in Fig.5, each channel in the polychromator is output to an independent PMT. How the authors deal with the differential aging of the the N2 and H20 PMTs? Since the ratio of the two signal is used to calculate the mixing ratio, a drift in gain or sensitivity of the PMT of one channel will not necessarily match nor correspond to a possible drift of the other PMT. This is a well-known problem in literature, and different groups apply different solutions. could you comment on that?

---

## Author Response (AR1)

Reply to report #1 by referee #2

The paper by Totems et al. is a comprehensive study on different bias sources in Raman lidar measurements of temperature and water vapor mixing ratio, including mitigation strategies. The paper is not only interesting for this type of tropospheric lidar, but some aspects can be transferred to all other types of lidars. For instance, the spatial inhomogeneities of PMT sensitivity or questions of range-dependent beam overlap can be crucial for other lidars, too. After a general description of the potential bias sources and their mitigation (or at least quantification), the authors apply this information to their own Weather & Aerosol Raman lidar WALI. The authors document that WALI fulfills the criteria set by WMO.

The manuscript is mostly well written, conclusive and well elaborated. The general methods of identification of potential bias sources are extensively described, even if some points are still confusing. Furthermore, I am missing some details in the application of these methods to WALI. Here, a more careful description is needed. Further details are given below.

**We thank the referee for a very thorough and helpful review. Our replies are detailed below.**

Specific comments:

l. 113: It is not clear to me, why r^ and T^ are used here, instead of r and T used in the remaining manuscript.

**We meant to make a difference between the actual values r and T, and their estimates given by the lidar for which the customary r^ and T^ are used. The latter being subject to biases r^-r, T^-T. We have clarified this in the text: "(where ^ denotes an estimate, to be distinguished from the true value without ^)"**

l. 137: Do you expect significantly different errors at other temperatures?

**Since Q(T) is nearly linear but not exactly so, the difference is small but non-zero (~20% in a typical temperature profile). This is why we state that this value is taken at 0°C (arbitrarily, at the freezing point of water most relevant for weather models).**

l. 142/143: This sentence should be rephrased (because R' is not needed at all for temperature measurements). Maybe: In order to fulfill WMO requirements for temperature and WVMR measurements, the SNR of the Q' ratio must be 6-10 times larger than the R' SNR, respectively.

**We agree. But because this is not limited to SNR but also to any inaccuracy, we propose: "In order to fulfill WMO requirements for temperature and WVMR measurements, the Q' ratio must be 6-10 times more accurate than R'."**

l. 144-146: a) The phrasing seems to be odd. Please improve. b) If I understand correctly, the authors want to emphasize the relevance of bias estimation. However, they concentrate in the next sentences on SNR.

**We tried to improve the phrasing as such: "However, Raman cross-sections are larger for the RR channels than for the H2O VR channel. Hence when dealing with a RR+VR lidar rather than a VR system, the main difficulties are not only due to low signal-to-noise ratio (SNR) but also encompass strong constraints linked to instrumental biases.". It is true the rest of the section focuses on SNR constraints. We did not want to emphasize the importance of bias at the expense of random noise, both have the same impact on error, even though the latter cancels out with integration.**

Section 2.3: This section is partly only relevant for receivers using free-space optics, with the described topics being of minor importance, if fibers are used. I recommend making clearer, which part of this section is of general relevance, because applicable to all kinds of receivers, and which is only relevant for receivers without fiber.

**We are unsure why the referee would deem necessary to discriminate here between fibered receivers and receivers using free-space optics. Indeed the former also use free-space optics before a fibered element. The WALI also includes a fiber, which we have proven not to mitigate the bias sources (cf. section 3.2.1). Due to our uncertainty about this comment from the referee, and our position that the described sources of bias affect any type of receiver, we have elected to keep this section as it was.**

l. 229-231: This sentence sounds odd. Please rephrase.

**We propose the following for better clarity: "Range-dependent biases influence the lower part of the lidar profiles exactly like the overlap factors. They significantly impact the profile up to a given range from the emitter, depending on the characteristics of the receiving optics as seen above, but also on the quality of the alignments, which is seldom twice the same. Two methods are used in the literature to estimate the actual overlap factors of a Raman lidar:"**

l. 257: Please explain why a range-averaging should be applied here to calculated a range- dependent overlap ratio.

**To reach sufficient SNR and have the required per mil precision on bias, range-smoothing (and indeed not averaging, thanks for pointing it out) is necessary on top of time-averaging. We use a gradually larger smoothing kernel with range (a few meters below 200 m to a few hundred meters above 1 km). We put in the text: "the overlap ratios can be estimated with suitable precision (~$10^{-3}$) by averaging the signals over time and smoothing them over range."**

The procedures with horizontal and vertical beam seem to be identical, but only the differential extinction becomes more important with vertical beam. It is confusing to repeat the equations 19 and 20 (for horizontal beams) as 21 and 22 (for vertical beams), but introduce at the same time implicitly the overlap ratio OR. I suggest to define OR explicitly, but remove the Eq. 21 and 22.

OR is partly used with a "hatch" (l. 104) and partly not (l. 581), but the difference does not become clear to me.

**There is indeed a case to be made for clarity here. For instance, Eqs. 19 to 22 are all for horizontal beams. We first considered following the suggestion of the reviewer to remove Eqs. (21) and (22), which gave the following:**

**"If for instance the lidar can be mounted on a rotating platform capable of aiming horizontally, the overlap ratios OR_R = O_H2O/O_N2 and OR_Q = O_RR2/O_RR1 can be estimated with suitable precision (~10-3) by averaging the signals over time and smoothing over range, and finally correcting for differential of extinction on the VR ratio. These estimates of OR_R and OR_Q will then be used during signal processing in Eqs. (4) and (5)."**

**In this case, how the estimates are calculated seems too implicit in our opinion, whereas they play a very important role in debiasing the measurements. Instead we propose to be more explicit even if redundant, with the following text:**

**"If for instance the lidar can be mounted on a rotating platform capable of aiming horizontally, the overlap ratios OR_R = O_H2O/O_N2 and OR_Q = O_RR2/O_RR1 can be estimated with suitable precision (~10-3) by averaging the signals over time and smoothing them over range, and finally correcting for differential of extinction on the VR ratio:**

**Eq (21) - Eq (22)**

**These estimates of the overlap ratios will then be used during signal processing for vertical shots as in Eq. (4) and (5)."**

**As now explained in the beginning of section 2, x with a hatch denotes the estimated value of variable x.**

Please explain additionally the difference between R(z) with and without overbar.

**As explained below Eq. (16), x with overbar denotes the expected value of variable x. Without a bar, there would be a noise term to add in Eqs. (19) and (20).**

l. 271/272: If the overlap measurement has only limited relevance in practice, this should be emphasized already in the beginning of this section.

**We have changed the wording in the beginning of the section "Two methods are used in the literature to APPROXIMATE the actual overlap factors of a Raman lidar:". We believe that adding more at that point would be confusing, and that the ending paragraph "These difficulties make it extremely challenging to estimate the overlap ratios with an accuracy better than a few percent" is enough to explain the problem.**

Furthermore, I suggest adding a comparison of capabilities of the "theoretical method" described in Section 2.4 and the empirical correction (practical method) described later.

**We think the difference between these capabilities is already shown in practice in Figure 12 b) e) by the corrections functions found necessary to correlate best with in-situ measurements.**

**Incidentally, we have tested a new method in the past weeks using a very large plane mirror on a carefully balanced support system to reflect all channels horizontally, instead of tilting the whole lidar. This seemed to largely alleviate the need for a further empirical correction.**

l. 289-291: This seems to be a contradiction to the statement in l. 361. Please explain.

**Indeed, the fiber did not help with range-dependent biases, contrary to what we initially thought. We have clarified this by using a hypothetical: "Fiber optics are also known to partly scramble the input illumination, which COULD HELP MINIMIZE the range-dependence of filter transmittance or detector sensitivity for the different Raman channels."**

l. 303: Could you please explain the "build-in leaks at 532 nm"?

**Leaks of green visible light at 532 nm, from a pair of dichroic plates at the output of the commercial Quantel laser that were supposed to filter out the fundamental and secondary harmonics, are still sufficient to be the constraining parameter for eye safety. We tried to clarify this sentence: "as limited by leaks at 532 nm through the built-in filtering dichroic plates".**

l. 316-318: If I understand correctly, you expect a 4°C change of the seeder temperature within 5 min. This seems to be quite large, even for an uncontrolled system. Please check. Does it run into a more stable state after a few hours of operation?

**We understood much later with the help of colleagues who owned the same seeder (unfortunately not from Quantel who was very reluctant to comment on seeder stability) that the seeder requires several hours of pumped operation before being properly stable. In that case, we had just turned it on for a few minutes. This, as well as the rudimentary construction of our interferometer, not thermally stabilized either, with a large estimated measurement error (~0.1 pm), explains the observed residual drift. That is also why we only provide an upper bound for the wavelength stability. We added a few words to clarify this in the paragraph.**

l. 358-361: This is an interesting result, but double negations should be avoided. I suggest writing: "... even with the use of fiber optics the angle of incidence on the interference filters depends on the image positions in the focal plane of the telescope (i.e. mainly the distance to the optical axis), in contrast to what could be expected."

**Thank you for this proposition, we have amended the text as suggested.**

l. 361: As mentioned above, this seems to be a contradiction to the statement in l. 290.

**We have proposed a solution hereabove.**

l. 446: Line 332 says that a 1 mm fiber is used for soundings and Line 433 says that a 0.6 mm fiber is used for alignment/tests. Please check for consistency.

**There are two steps. First, for the alignment itself, we use narrower fibers (600μm input, 200μm output) to better constrain the alignment. Then, for qualification, we use the actual fiber of WALI (1000 μm) as an input, and the 600 μm fiber to feed the OSA. We have re-read the text and believe it to be sufficiently clear.**

Figure 8: Please explain the normalization of the sensitivity. It seems odd that it is nearly always above 100%.

**Indeed, the normalization is relative to the central value. We have clarified this in the figure caption: "Sensitivity is given normalized by its value at the approximate mechanical center of the PMT or at normal incidence as determined using the reflection on the attached neutral density filter."**

l. 477: In l. 464 a diameter of 1 mm is given. Please check for consistency.

l. 483: In l. 479 a diameter of 21 mm is given. Please check for consistency.

**These mistakes have been corrected: 1 mm and 22 mm are the true values. Thank you for pointing them out.**

l. 486: Could you please give an example, which effect this sensitivity change has on the result (i.e. WVMR or T profiles)?

**We have added: "Using the calculations in Section 2, a $\theta = 1$ mrad field angle corresponds to 0.39° incidence on the PMT, inducing potentially 0.8-2% bias on R and Q, and thus a significant 1 to 2°C bias on temperature."**

l. 496: It would be interesting to see an example of these spikes and to learn how their effects are mitigated.

**As we do not record single shots, we unfortunately do not have examples of these to show here. We only see them on the live night-time shot-to-shot signal of the lidar on our Labview interface. They are intermittent spikes, probably from the flashlamp or Q-switch pulses of the laser, appearing periodically at ~7, 10, 13, 16, 19 km. We have yet to completely understand their cause and how to reliably mitigate them. But since this is a very technical, empirical, and partly unresolved aspect, we preferred not to detail it more in this paper. The method we found for mitigation is broadly explained after line 498.**

l. 508: If I understand correctly, the baseline varies with time and is therefore measured every 8 min. Is this baseline subtracted from the previous or the following profile series? How strong is the variation with time?

**We have added a sentence to answer these questions: "Note that baseline variation is not significant between successive evaluations without an external perturbation; the estimated baseline is automatically subtracted from the profiles before recording during the next 8 minutes (Eq. (3))."**

l. 570: I recommend avoiding highly subjective terms like "rather lukewarm".

**Sorry for this oversight. The new sentence reads: "The overlap factors and their ratios were estimated on signals averaged over 3 hours after sunset on December 19th, 2019, with a tepid (14°C), non-turbulent but hazy atmosphere".**

l. 605: The remaining bias of OR seems to be of similar magnitude like the initial correction. Does this make the elaborate overlap measurement with horizontal beams obsolete?

**The comparison should be made between Figure 11 and Figure 12. We see for temperature that the magnitude of the secondary correction is actually smaller than the first. For water vapor, they are indeed of similar magnitude.**

**It was still indispensable in our opinion to perform the study with a horizontal system, so as to ascertain without a doubt that biases are found in a homogeneous atmosphere.**

**As said earlier, we are now using a large plane mirror on a special support system to reflect all channels horizontally, instead of tilting the whole lidar. With no changing mechanical constraints on the emission and reception channels, this largely reduced the secondary correction function. The remaining difference can now be explained by horizontal inhomogeneity in the atmosphere.**

Figure 12 b/e and l. 608: The displayed data shows a very large scatter. Please provide also the error of the mean. Please explain the model to get the correction for OR (red line) from the measured mean ratio.

**We have added the error for the mean (standard deviation divided by the square root of the number of samples) as vertical bars on the b) and e) graphs. The models are a sum of three exponential falls with coefficients, as explained in the added sentence: "We fit a sum of three exponential falls to the mean, of the form: $1 + a1 \exp(-z/z1) + a2 \exp(-z/z2) + a3 \exp(-z/z3)$, with $ai$ coefficients and $zi$ ranges to be adjusted."**

l. 643/644: What kind of variability do you mean? Spatial or temporal? Why does it only affect nighttime-data?

**We have added precisions in this sentence to try and answer these questions: "On WVMR, the results show little bias, and RMS deviation is dominated by spatial atmospheric variability at night and at low altitude (when SNR is high), and by lidar noise in all other cases".**

l. 646: I wonder why a wrong correction of OR shall be responsible here. OR has initially been measured and then corrected using the same set of radiosondes that is now used for comparison. Please comment.

**This could be due in part to an underestimated correction of ORQ (only ~0.8%) due to the regularized model used to approximate it, but also to local effects.**

Regarding this whole section, I would prefer to see an independent comparison with another set of radiosondes than used for the corrections and calibrations.

**This is for sure better practice. It was however difficult in fact to calibrate the Raman channels and especially their corrections on a smaller set of radiosondes, and after that period of very good weather, a spell of bad weather made more comparisons unfeasible (dense cloud cover after June 3rd, until June 22nd). The system was turned off. After such a long time, another type of bias due to long-term drifts could be expected (they were measured in August 2020, the correction functions remain the same, but the calibration curves/coefficients changed); this is an interesting matter, but we deemed this to be outside the scope of this paper. Here we preferred to show the range-dependent errors in case of a time-concomitant calibration, which are still non-zero. We believe this dataset allows that kind of study.**

l. 653: Please provide the distances between lidar/radiosonde and their respective closest ERA5 gridpoint, and between these two gridpoints.

**In the attached figure, we provide a map showing the position of the RS launching station at Trappes, the lidar at LSCE, the ERA5 grid points and "pixel" limits, and the RS trajectories up to 6 km. We hesitated about adding this as an appendix, but it does not seem necessary to make such a large addition to the paper and it would take too much time to make it a proper copyright-free map.**

Has the drift of the radiosonde been taken into account or is the drift during the ascent up to 6 km much smaller than the distance between LCSE and Trappes sites (and ERA5 gridpoints)?

**According to ERA5 reanalyses, the ground-level wind was mostly from the NE (except a few occurrences of NW and SW winds) and at most 10-15 km/h for this period. This led a lot of the radiosondes to drift away from both Trappes and LSCE, yet a majority remained in the "pixel" of Trappes (see attached figure).**

**Given that our need was for an estimation of the horizontal variability of the fields between lidar and radiosondes, we feel that the variability from one grid point to the next (yellow arrow), is a good representation for this. We did not go so far as to follow the drift of the radiosondes.**

[Figure]

**In the table caption, we have added: "The grid points are located 8 km WNW of the lidar and 2 km S of the RS station respectively, 18.3 km apart, and almost all RS trajectories below 6 km altitude are contained within the same "pixel" of the ERA5 fields."**

l. 667/668: It would be interesting to learn about potential reasons for these remaining differences, even if the authors cannot verify these in the context of this paper.

**We propose the following text to complement the analysis, and better conclude our study:**

**"There is still a discrepancy with the mean difference of temperature however; it is not fully explained by the differences of temperature at the locations of the two profiles seen in ERA5. The model used to approximate the correction may be imperfect, and introduce small errors when the necessary correction is large and fast-varying. We aim to improve this in the future by a better estimation the overlap ratios horizontally, for instance using a large folding mirror instead of tilting the lidar, which induces varying mechanical constraints on the optics. This should greatly reduce the necessary correction function and the remaining error should be limited by the horizontal inhomogeneity of the atmospheric temperature over a few kilometers."**

Technical comments / typos:

l. 19: typo "homogeneous"

l. 167: Please add: "0.12-0.4 % for temperature"

l. 171: Please introduce the sigma variable

l. 187: typo "thus"

l. 193: "effective refractive index"

l. 403: OD is not given in the figure.

l. 466: Add "neutral density filter"

l. 474: "not vary by more ..."

l. 484: "convolution with ..."

l. 525: typo "leads"

l. 577: typo "channels"

Fig. 11: The "^" in the legend should be removed.

l. 770 and 779: The superscript "2" should be corrected, here.

Reply to report #2 by referee #1

General comments:

The study "Mitigation of bias sources for atmospheric temperature and humidity in the mobile Weather & Aerosol Raman Lidar (WALI)" by Julien Totems, Patrick Chazette and Alexandre Baron provides a thorough description of the WALI system both from the point of view of the technical characterization of the lidar transceiver and the performances in terms of bias and RMS. The article is well written, easy to read and exhaustive. All topics are described and supported by either previous literature or statistical studies performed by the authors. In Sect. 4.3, the comparison with the radiosounding, in addition to calibration purposes, brings useful information about the quality of the WVMR and temperature data. Accepting the 12 km distance between lidar and in-situ measurements, and then accepting a higher RMS due to slightly different atmosphere, it allows to use the bias and RMS values as solid evaluation of the WALI performance. There are only few technical comments/remarks that should be addressed before publication. I strongly recommend this manuscript to be published in AMT.

**We thank the referee for this very encouraging review and their useful comments.**

Technical comments:

Abstract: while the abstract sets the main objectives of the study within the state of the art, it does not state any quantitative result. In this way, the abstract fails to deliver to the reader a concise summary of the obtained results. The main results regarding calibrations and comparisons with radiosounding presented in section 4 should be reported in the abstract by stating the mean daytime and nighttime biases and RMS values.

**We have added the following results to end of the abstract: "For temperature, the magnitude of the highlighted biases can be much larger than the targeted absolute accuracy of 1°C defined by the WMO (up to 6°C bias at low range). Measurement errors are quantified using simulations and a number of radiosoundings launched close to the laboratory. After de-biasing, the remaining mean differences are below 0.1 g/kg on water vapor, 1°C on temperature, and RMS differences are consistent with the expected error from lidar noise, calibration uncertainty, and horizontal inhomogeneities of the atmosphere between the lidar and radiosondes."**

Introduction, ln 55, 64, 71, 76: the authors mention the "sources of biases". A statistical bias is typically a systematic error, a difference between the measurement and the truth. In this sense it would be more appropriate to refer to the "sources of uncertainty" or "sources of error".

**We have replaced bias by systematic error in 2 occurences, and the last by causes of bias.**

Sect.2.1, Pg 4, ln 90-91: which are the "required altitude and time"?

**We chose not to discuss the integration parameters here so as to reserve this section for theoretical considerations that could be applied to any lidar. These values are given in the beginning of section 4.3.**

Sect.2.1, Pg 4, ln 101: "corrected for"

**This has been corrected.**

Sect.2.1, Pg 4, ln 106-109: have the authors actually did some simulation to assess the reliability of the 5%-impact of the differential extinction or the estimate by Whiteman 2003 is taken directly?

**As stated below, we use an average atmospheric density profile to compute DeltaTau(z). Whiteman 2003 does not estimate it, but rather shows this method to be efficient, because aerosols do not interfere much with the result if their effect is approximated by the Raman-derived optical thickness and an average Angström exponent (~1).**

Sect.2.1, Pg 4, ln 110-111: do the authors mean that the N2 has a constant mixing ratio through troposphere and stratosphere? The statement is not formulated in a clear way.

**We have tried to clarify the sentence as suggested.**

Sect.2.2, Pg 5, ln. 126-129: the authors set the requirement for successful monitoring, verification and data assimilation into models by listing noise errors and biases. If I interpret correctly what the authors mean by bias, this should not be part of the requirement as they can be efficiently removed by the calibration process.

**This is an overarching yet important question. Biases can be calibrated, but it must be done using a reference, which may be imperfect, and this calibration can become obsolete if the biases vary in time. It is preferable to mitigate them in the first place, and even then monitor regularly that they are kept within acceptable bounds when the measurement is compared to an internationally recognized reference. Therefore, the size of this window, what we call requirements, around the true values, is of prime importance.**

Sect. 2.3, pg 8, ln 187: "thus"

**The correction has been made.**

Sect. 2.3 pg 9, Figure 1b: the caption does not say what the green lines represent. One can imagine that is the return beam from the IF, but it is not clear.

**We have added in the caption: "Green/red/blue lines represent rays from infinity/finite distance/offset emitted beam, respectively."**

Sect. 3.3,pg 17 ln 398: what material the cage system is made of? Is the cage subject to thermal expansion?

**We use the widely available 6 mm steel rod cage system distributed for instance by Thorlabs. It is indeed subject to thermal expansion, but the whole structure is thermally regulated (metal-to-metal contact with a TEC cooled plate), and enclosed in a well-adjusted 3D-printed plastic box for insulation. The temperature measured by a sensor set in one of the optical filter mount is typically kept constant within a few $10^{-1}$°C in our air-conditioned laboratory.**

Sect. 3.3-3.4: the authors perform a thorough analysis of the detectors' sensitivity, calibration and responses. PMT sensitivity and gain are also analysed in detail, which allows correcting for inequalities at the PMT output. As it is shown in Fig.5, each channel in the polychromator is output to an independent PMT. How the authors deal with the differential aging of the N2 and H20 PMTs? Since the ratio of the two signal is used to calculate the mixing ratio, a drift in gain or sensitivity of the PMT of one channel will not necessarily match nor correspond to a possible drift of the other PMT. This is a well-known problem in literature, and different groups apply different solutions. Could you comment on that?

Indeed we have observed this phenomenon as well, and that is why both WVMR and temperature Raman channels have to be recalibrated every time we put our system in operation. Between June $2^{nd}$ and August $2^{nd}$, 2020, we have observed a 7.3% change of the water vapor calibration coefficient, and 2.1°C offset of the temperature calibration curve; however in that case a failure of the laser seeder (bi-modal behavior and central wavelength change) is mostly to blame.

In this already dense paper, we choose not to address the temporal aspect of biases, i.e. the evolution of calibration due to several causes among which PMT aging, because it is a wide subject. During the period studied here, spanning only two weeks, no such effect was consistently highlighted. We will definitely have to address it in the future, when WALI is involved in a long campaign like WaLiNeAs (2-3 months). Regular dedicated radiosondes will be launched directly from the lidar site to allow a maintained calibration.

We have added the following comments as perspectives in the conclusion: "The long-term temporal evolution of Raman channel calibration, expected from various effects like differential PMT aging or laser seeder drift, induces biases variable in time over the time-scale of such a project (several months). This aspect is becoming a main focus as the community works towards operational uses of weather Raman lidars (eg. Hicks-Jalali et al., 2020)."